# DyMoDreamer: World Modeling with Dynamic Modulation

**Boxuan Zhang[1], Runqing Wang[1], Wei Xiao[1], Weipu Zhang[1],**
**Jian Sun[1], Gao Huang[2], Jie Chen[1], Gang Wang[1]***

[1]School of Automation, Beijing Institute of Technology
[2]Department of Automation, BNRist, Tsinghua University

## Abstract

A critical bottleneck in deep reinforcement learning (DRL) is sample inefficiency, as training high-performance agents often demands extensive environmental interactions. Model-based reinforcement learning (MBRL) mitigates this by building world models that simulate environmental dynamics and generate synthetic experience, improving sample efficiency. However, conventional world models process observations holistically, failing to decouple dynamic objects and temporal features from static backgrounds. This approach is computationally inefficient, especially for visual tasks where dynamic objects significantly influence rewards and decision-making performance. To address this, we introduce DyMoDreamer, a novel MBRL algorithm that incorporates a dynamic modulation mechanism to improve the extraction of dynamic features and enrich the temporal information. DyMoDreamer employs differential observations derived from a novel inter-frame differencing mask, explicitly encoding object-level motion cues and temporal dynamics. Dynamic modulation is modeled as stochastic categorical distributions and integrated into a recurrent state-space model (RSSM), enhancing the model's focus on reward-relevant dynamics. Experiments demonstrate that DyMoDreamer sets a new state-of-the-art on the Atari 100k benchmark with a $156.6\%$ mean human-normalized score, establishes a new record of $832$ on the DeepMind Visual Control Suite, and gains a $9.5\%$ performance improvement after 1M steps on the Crafter benchmark. Our code is released at https://github.com/Ultraman-Tiga1/DyMoDreamer.

## 1 Introduction

Deep reinforcement learning (DRL) has achieved significant breakthroughs in sequential decision-making tasks [1, 2, 3, 4]. State-of-the-art methods such as DQN [5], PPO [6], MuZero [7], and EfficientZero [8] have demonstrated impressive performance across various benchmarks. However, these methods often suffer from low sample efficiency, requiring hundreds of millions of interactions with the environment [9], thus limiting their practicality for real-world applications where data collection is costly or infeasible. A key approach to addressing this challenge is the general framework of world models, which learns compact latent dynamics for planning and decision-making. One early instantiation of this framework [10], combines a VAE with an RNN and uses evolutionary strategies to optimize policies in the learned latent space. This framework combines a VAE with a recurrent neural network (RNN) and uses evolutionary strategies to optimize the policy within a latent space. By generating synthetic interaction data or "imagination trajectories" [11, 12], world models allow agents to learn behaviors entirely within the learned model. This model-based approach reduces the reliance on direct environment interactions, significantly improving sample efficiency.

---

*Corresponding author: `gangwang@bit.edu.cn`.

39th Conference on Neural Information Processing Systems (NeurIPS 2025).

Traditionally, RNNs have been employed to capture temporal dependencies in world models. For instance, the Dreamer series (DreamerV1-3) [11, 13, 14] has demonstrated human-level performance on challenging benchmarks like Atari 100k [15] and Minecraft [16] using the recurrent state space model (RSSM) [17]. More recently, transformer-based world model architectures [18] have revolutionized sequence modeling, introducing methods such as GPT-like models [19] and Transformer-XL [20], and spatio-temporal transformers [21]. These advances have further refined the capabilities of world models through tokenization techniques [9, 22, 23].

Despite their success, both RSSM- and transformer-based world model architectures face challenges in accurately modeling small dynamic objects [24]. VAEs tend to amplify randomness, disproportionately affecting the parameters of temporal models [25]. Moreover, small dynamic objects often carry a higher degree of decision relevance compared to static environment backgrounds [26]. For example, in Atari's game *Pong*, the paddle and ball are critical for decision-making, while the static red background is irrelevant. Recent techniques such as IRIS [9, 22, 23] and DIAMOND [24] partially mitigate these issues by improving the precision of image reconstruction in world modeling, but at the cost of significant computational overhead. Similarly, the recent approach OC-STORM [21] integrates vision models with world models, and introduces segmentation masks for object annotation, which require extensive pretraining and prior knowledge of object counts.

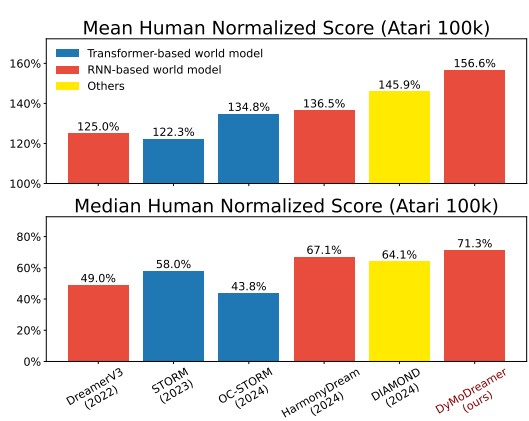

Figure 1: DyMoDreamer surpasses existing state-of-the-art model-based methods.

In this paper, we introduce DyMoDreamer, a novel world model architecture for model-based RL that improves dynamic feature extraction and enriches temporal information by encoding differential observations. DyMoDreamer integrates a dynamic modulation mechanism into RSSM, emphasizing critical environmental variations without relying on high-precision image reconstructions or prior object annotations. Through a novel lightweight inter-frame differencing mask, our approach effectively prioritizes decision-relevant dynamic features, addressing key limitations of existing RNN- and transformer-based models. Our approach is inspired by the cognitive processes observed in human infants, who naturally focus on dynamic object interactions to infer fundamental principles about their surroundings [27, 28]. Leveraging this principle, DyMoDreamer achieves superior policy performance, demonstrating that emphasizing dynamic objects and temporal information significantly improves decision-making in RL. Our method establishes a new SOTA of the RNN-based world model with $156.6\%$ human-normalized score on the Atari 100K benchmark, sets a new record with 832 mean score on the DeepMind Visual Control Suite, and delivers a $9.5\%$ performance improvement on the Crafter benchmark after 1M environment steps.

Our main contributions are summarized as follows:

- We introduce a dynamic modulation mechanism that enhances RSSM by encoding differential observations to focus on key dynamic features and temporal information.

- We develop DyMoDreamer, a novel world model that integrates dynamic modulation into recurrent states, improving policy efficiency by prioritizing dynamic environmental features.

- We achieve state-of-the-art performance of RNN-based world model with $156.6\%$ human-normalized score on the Atari 100k benchmark, 832 average score on DeepMind Visual Control, and $9.5\%$ performance improvement on the Crafter benchmark within 1M steps.

The remainder of this paper is structured as follows. Section 2 details our methodology. Section 3 presents empirical evaluations on the Atari 100k benchmark, the DeepMind Visual Control Suite, and the Crafter benchmark, followed by ablation studies in Section 4. Detailed scores and curves are provided in Appendix E, alongside extensive ablation studies in Appendix J. We conclude with insights and future directions in Section 6.

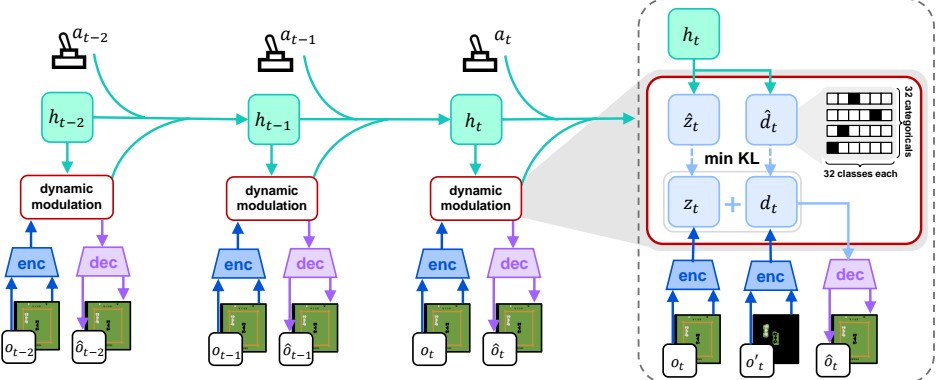

Figure 2: Overview of DyMoDreamer. The model integrates dynamic modulation derived from differential observations to enhance the perception of dynamic features within the environment.

## 2 Methodology

This section outlines DyMoDreamer, a model-based reinforcement learning algorithm designed to enhance dynamic feature extraction for improved agent performance. The problem is formulated as a partially observable Markov decision process (POMDP), represented by the tuple $(\mathcal{O}, \mathcal{A}, \mathcal{T}, R, \gamma)$, where $o_t \in \mathcal{O}$ denotes high-dimensional image observations, $a_t \in \mathcal{A}$ represents discrete actions generated by some policy $\pi(a_t \mid o_{1:t}, a_{1:t-1})$, $\mathcal{T}$ is the transition function, $R$ is the reward function associated with a particular task, and $\gamma$ is the discounting factor. The objective is to learn a policy $\pi$ that maximizes the expected sum of discounted rewards $\mathbb{E}_\pi \left[ \sum_{t=1}^{\infty} \gamma^{t-1} r_t \right]$, where $r_t = R(o_{t-1}, a_{t-1})$.

We present DyMoDreamer, a model-based RL framework designed to improve world model representations of dynamic temporal features and enhance agent performance. At its core, DyMoDreamer employs the RSSM architecture, augmented with novel dynamic modulation mechanisms that selectively emphasize dynamic features and temporal information in the environment. This approach draws inspiration from advances in modulated ordinary differential equations (ODEs) [29], where independently trained modulators improve temporal modeling for long-term prediction in complex systems. Moreover, the latent flow method [30] in model free reinforcement learning (MFRL) suggests that capturing temporal changes via time-based differencing can enhance transition modeling and improve the performance [31]. Building on these insights, DyMoDreamer introduces a temporal frame differencing method to refine the expressiveness of dynamic modulators, enabling the agent to leverage richer and more decision-relevant information for improved policy learning and performance.

We mainly base our approach on DreamerV3 [14], introducing dynamic modulation into the RSSM world model architecture. Below, we detail the construction of inter-frame differencing masks for dynamic modulation (Section 2.1), the integration of dynamic modulation into the world model (Section 2.2), and the end-to-end training process (Section 2.4).

### 2.1 Dynamic Modulation

As a core component of the world model, we leverage a VAE with convolutional neural networks (CNNs) to encode observations into stochastic state representations. The original encoder in DreamerV3 integrates temporal information, modeled by the hidden states $h_t$ (introduced in Section 2.2 along with RSSM), and maps observations $o_t$ to stochastic representations $z_t$. To focus on the dynamic aspects of the environment, we introduce dynamic modulation, which explicitly captures temporal changes between frames.

Differential observations $o'_t$ are derived by computing inter-frame differences from the original observations, effectively isolating dynamic components. A dedicated dynamic encoder processes $o'_t$ to generate dynamic modulators $d_t$, which participate in the reconstruction of $o_t$ to ensure the

accurate extraction of dynamic features. The encoding process is as follows:

$$
\begin{aligned}
\text{Stochastic encoder:} \quad & z_t \sim q_\phi\big(z_t \mid h_t, o_t\big) \\
\text{Dynamic encoder:} \quad & d_t \sim q_\phi\big(d_t \mid h_t, o'_t\big) \\
\text{Decoder:} \quad & \hat{o}_t = p_\phi\big(\hat{o}_t \mid h_t, z_t, d_t\big).
\end{aligned}
\tag{1}
$$

The primary claim of this work is that explicitly integrating dynamic modulators, which capture dynamic environmental features, significantly enhances agent performance. By modeling features through differential signals, the approach improves the extraction of critical dynamic patterns and bolsters overall robustness [32]. To ensure the dynamic modulators emphasize decision-relevant patterns in the environment, we construct differential observations $o'_t$ using a frame differencing method. Since forward differencing requires access to future information, we instead define a temporal backward differential binary function:

$$
D(o_{i,h,w,c}, o_{i-k,h,w,c}) =
\begin{cases}
1, & \text{if } \|o_{i,h,w,c} - o_{i-k,h,w,c}\|_2 > \epsilon \\
0, & \text{others}
\end{cases}
\tag{2}
$$

where $\epsilon$ is a binary threshold value, set to $0.001$ and $k$ is an integer hyperparameter represents the differenced frames' interval (empirically set to 1, increasing $k$ helps suppress potential static background flash), $i$ is the time dimension, $h$ and $w$ are the spatial dimensions, and $c$ is the channel dimension. In fact, very few pixels may exceed the threshold $\epsilon$ after processing by $D(\cdot)$. To address this sparsity, we expand the regions of interest by filling the 0-pixel regions surrounding 1-pixel regions with 1's, a process achieved using convolution operations. This yields dynamic differential masks $M(o_t)$, which emphasize both the differing pixels between frames and their surrounding regions. Subsequently, the differential observations are defined as the masked original observations $o'_t = M(o_t) \cdot o_t$ and are utilized to determine the dynamic modulators. This design explicitly extracts dynamic features from the observations, effectively capturing temporal changes rather than merely amplifying the influence of the most recent frame in the recurrent information. Unlike approaches that rely on sophisticated vision techniques, computing differential observations imposes negligible computational overhead while yielding notable performance gains (Section 3.2). Section 2.3 discusses the motivation for performing differencing in the observation space rather than in the latent space. Furthermore, additional results in Appendix G demonstrate that DyMoDreamer can flexibly adapt to different differencing strategies tailored to varying task environments. Differential observations across different benchmarks are provided in Appendix F.

## 2.2 Dynamically Modulated World Model

During interaction with the environment, DyMoDreamer collects observations, actions, and rewards into an experience dataset. After a predefined number of interactions, trajectories are sampled from this dataset to train both the world model and the categorical VAEs [33, 34]. The world model is built on RSSM, where the sequence model utilizes the recurrent latent state $h_t$, to predict the sequence of stochastic representations and dynamic modulators based on past actions $a_{t-1}$. The model state is formed by concatenating $h_t$, $z_t$ and $d_t$, enabling the prediction of rewards $\hat{r}_t$ and episode continuation flags $\hat{c}_t \in \{0, 1\}$:

$$
\begin{aligned}
\text{Sequence model:} \quad & h_t = f_\phi\big(h_{t-1}, z_{t-1}, d_{t-1}, a_{t-1}\big) \\
\text{Latent predictor:} \quad & \hat{z}_t = p_\phi\big(\hat{z}_t \mid h_t\big) \\
\text{Dynamics predictor:} \quad & \hat{d}_t = p_\phi\big(\hat{d}_t \mid h_t\big) \\
\text{Reward predictor:} \quad & \hat{r}_t = p_\phi\big(\hat{r}_t \mid h_t, z_t, d_t\big) \\
\text{Continue predictor:} \quad & \hat{c}_t = p_\phi\big(\hat{c}_t \mid h_t, z_t, d_t\big).
\end{aligned}
\tag{3}
$$

Since directly modeling the dynamics from raw images is computationally expensive and prone to errors, we convert the observation $o_t$ into latent stochastic categorical distributions $q_\phi\left(z_t \mid h_t, o_t\right) = \mathcal{Z}_t$ for the latent variable $z_t$, which consists of 32 categories, each with 32 classes. Similarly, the dynamic modulator $d_t$ is sampled from $\mathcal{D}_t = q_\phi\left(d_t \mid h_t, o_t\right)$, where $\mathcal{D}_t$ also consists of 32 categories, each with 32 classes:

$$
z_t \sim q_\phi\big(z_t \mid h_t, o_t\big) = \mathcal{Z}_t, \quad d_t \sim q_\phi\big(d_t \mid h_t, o'_t\big) = \mathcal{D}_t.
\tag{4}
$$

We apply the straight-through gradients trick [35] to maintain the gradients of representations and modulators, as sampling from a distribution inherently lacks gradients for backpropagation. The sequence model $f_\phi$ takes as input the hidden state $h_{t-1}$, the stochastic representation $z_{t-1}$, the dynamic modulator $d_{t-1}$, and the action $a_{t-1}$ from the previous step. Using this input, it computes a deterministic hidden state $h_t$ via a gated recurrent unit (GRU) [36]. Multi-layer perceptrons (MLPs) then utilize $h_t$, $z_t$ and $d_t$ to predict the current reward $\hat{r}_t$, the continuation flag $\hat{c}_t$, the dynamic modulation distribution $\mathcal{D}_t$, and the stochastic distribution $\mathcal{Z}_t$, respectively.

By design, the dynamic modulator $d_t$ is time-varying, capturing dynamic features that enhance the model's focus on decision-related dynamic objects and their surroundings in $o_t$. As a result, our world model demonstrates the following advantageous properties:

- *Dynamic feature embedding:* The dynamic modulator embeds dynamic features into the prediction models, compelling the model to prioritize decision-critical dynamic objects.

- *Temporal information capturing:* The dynamic modulator $d_t$, predicted by the hidden state $h_t$, enables the world model to capture temporal information across time sequences.

- *End-to-end joint training:* The dynamic modulator integrates into the reconstruction process alongside the stochastic representations and hidden states, facilitating end-to-end joint training of all world model components without requiring separate modulator training.

## 2.3  Intuition Behind Dynamic Modulation

In policy learning, rewards are predominantly influenced by the dynamic parts of observations and their adjacent static parts. For example, in the *Boxing* environment, the rewards are determined by effective jabs delivered by the agent, which involve dynamic elements (fists and arms) and their immediate static context (the body of the agent and its opponent) in the observations. Therefore, explicitly extracting dynamic local features in temporal sequences enhances the world model's ability to focus on decision-critical features, thereby improving policy performance. Moreover, since not all tasks rely exclusively on dynamic features for decision-making. For instance, in the *Gopher* environment, both dynamic elements (e.g., gophers and tunnels) and static objects (e.g., carrots to be protected) are critical for earning rewards. To address this, DyMoDreamer retains the original stochastic representations while augmenting them with dynamic modulators. We provide visualizations in Appendix H, which show that $z_t$ is indeed influenced by the dynamic modulation, leading to increased attention of $z_t$ to the static factors.

We choose to perform differencing in the observation space rather than in the latent space (as done in prior MFRL work [30], by adding $\delta_t = z_t - z_{t-1}$ to the sequence model), because the limited precision of latent encoding may hinder the detection of fine-grained dynamic features. For example, if a small but important object (such as the 1-pixel ball in *Pong*) is not captured in either $z_t$ or $z_{t-1}$, the latent difference would fail to reflect this motion. In contrast, frame-level differencing can make such movements explicit in the pixel space, and the dynamic modulation can incorporate this temporal information into modulation, enriching the dynamics modeling. Our ablation in Appendix J.3 further supports this motivation.

## 2.4  End-to-end Learning

In general, the parameters of the world model, denoted as $\phi$, are optimized end-to-end to minimize three types of losses: the prediction loss $\mathcal{L}_{\text{pred}}$, the dynamics loss $\mathcal{L}_{\text{dyn}}$, and the representation loss $\mathcal{L}_{\text{rep}}$. This optimization is performed in a self-supervised manner using a batch of sequences comprising observations $o_{1:T}$, actions $a_{1:T}$, rewards $r_{1:T}$, and continuation flags $c_{1:T}$. To further enhance the model's prediction performance, we introduce a differential divergence regularization term $\mathcal{L}_{\text{reg}}$, which leverages differential observations $o'_{1:T}$ to capture the dynamical differences between consecutive frames. The total loss function is defined in Equation (5), where $\omega_{\text{dyn}} = 0.5$ and $\omega_{\text{rep}} = 0.1$ represent the weight coefficients for the respective loss terms:

$$\mathcal{L}(\phi) = \mathbb{E}_\phi \Big[ \sum_{t=1}^{T} \big( \mathcal{L}_{\text{pred}}(\phi) + \omega_{\text{dyn}} \mathcal{L}_{\text{dyn}}(\phi) + \omega_{\text{rep}} \mathcal{L}_{\text{rep}}(\phi) \big) \Big] + \mathcal{L}_{\text{reg}}(\phi). \tag{5}$$

The prediction loss $\mathcal{L}_{\text{pred}}$ trains the categorical VAE to encode stochastic representations $z_t$ and dynamic modulators $d_t$ while reconstructing the input visual observations $o_t$. Additionally, it enables

the model to predict environment rewards and episode continuation flags, which are crucial for computing imagined trajectory returns during the behavior learning phase. The prediction loss comprises three components, as defined in Equation (6):

$$\mathcal{L}_{\text{pred}}(\phi) = \mathcal{L}_{\text{rec}}(\phi) + \mathcal{L}_{\text{rew}}(\phi) + \mathcal{L}_{\text{con}}(\phi) \tag{6}$$

where $\mathcal{L}_{\text{rec}}(\phi) = -\ln p_\phi(o_t|h_t, z_t, d_t)$ denotes the reconstruction loss of the input image, $\mathcal{L}_{\text{rew}}(\phi) = -\ln p_\phi(r_t|h_t, z_t, d_t)$ refers to the reward prediction loss, using a symlog two-hot loss, transforming the regression problem into a classification problem with soft targets, and $\mathcal{L}_{\text{con}}(\phi) = -\ln p_\phi(c_t|h_t, z_t, d_t)$ corresponds to the continuation flag prediction loss, which is implemented as a cross-entropy loss.

The dynamics loss $\mathcal{L}_{\text{dyn}}$ trains the sequence model to predict stochastic representations and dynamic modulators by minimizing the Kullback-Leibler (KL) divergence between the predicted and true distributions:

$$\mathcal{L}_{\text{dyn}}(\phi) = \max\left(1, \text{KL}[\text{sg}(q_\phi(z_t \mid h_t, o_t)) \parallel p_\phi(\hat{z}_t \mid h_t)]\right)$$
$$+ \underbrace{\max(1, \text{KL}[\text{sg}(q_\phi(d_t \mid h_t, o_t')) \parallel p_\phi(\hat{d}_t \mid h_t)])}_{\text{modulation's dynamic loss}}. \tag{7}$$

Here, $\text{sg}(\cdot)$ represents the stop-gradient operator, and the maximum operator clips the loss at 1 nat (approximately 1.44 bits) [37]. This "free-bit" operation prevents the loss terms from becoming excessively small, allowing the model to focus on other terms during training [38].

The representation loss $\mathcal{L}_{\text{rep}}$ enforces consistency between the priors and posteriors of $\mathcal{Z}_t$ and $\mathcal{D}_t$, allowing for a factorized dynamics predictor for efficient sampling during imagination training. It is defined as:

$$\mathcal{L}_{\text{rep}}(\phi) = \max\left(1, \text{KL}[q_\phi(z_t \mid h_t, o_t) \parallel \text{sg}(p_\phi(\hat{z}_t \mid h_t))]\right)$$
$$+ \underbrace{\max(1, \text{KL}[q_\phi(d_t \mid h_t, o_t') \parallel \text{sg}(p_\phi(\hat{d}_t \mid h_t))])}_{\text{modulation's representation loss}}. \tag{8}$$

The differential divergence regularization $\mathcal{L}_{\text{reg}}$ addresses the limitations of mean-squared error (MSE) loss, which primarily focuses on intra-frame differences. Instead, $\mathcal{L}_{\text{reg}}$ forces the model to learn inter-frame variations, improving its awareness of dynamic changes [39]. Backward differences are computed as $\Delta \hat{o}_t = \hat{o}_t - \hat{o}_{t-1}$ and $\Delta o_t = o_t - o_{t-1}$. These differences are transformed into probability distributions using the softmax function along the channel, height, and width dimensions:

$$\sigma(\Delta \hat{o}_t) = \frac{\exp(\Delta \hat{o}_t/\tau)}{\sum_{h,w,c} \exp(\Delta \hat{o}_t/\tau)}, \quad \sigma(\Delta o_t) = \frac{\exp(\Delta o_t/\tau)}{\sum_{h,w,c} \exp(\Delta o_t/\tau)}, \tag{9}$$

where $\tau$ is a temperature parameter (empirically set to $0.1$) to sharpen the distributions. The regularization term is then defined as the average KL divergence between these distributions across the batch $B$:

$$\mathcal{L}_{\text{reg}}(\phi) = \frac{1}{B} \sum \text{KL}\left(\sigma(\Delta \hat{o}) \parallel \sigma(\Delta o)\right). \tag{10}$$

## 2.5 Policy Learning

The policy in DyMoDreamer is trained exclusively on abstract trajectories predicted by the model using a standard actor-critic framework [40]. In our approach, the actor and critic operate on the concentration states $s_t = \{h_t, z_t, d_t\}$, leveraging the Markovian representations learned by the recurrent world model with dynamic modulation. Notably, due to the adoption of backward differential observations to encode the dynamic modulation, a random action is sampled for the initial frame to derive a single observation (or $k$ frames when $k > 1$) for starting the backward difference, resulting in $o_1'$. As we shown in Figure 3, although the reconstruction loss of imagined trajectories shows no significant reduction, we observe that DyMoDreamer generates substantially less hallucination on dynamic patterns during imagination compared to DreamerV3, which fundamentally explains the performance gains brought by our dynamic modulation mechanism. The remaining setup closely follows DreamerV3 and is detailed in Appendix D.

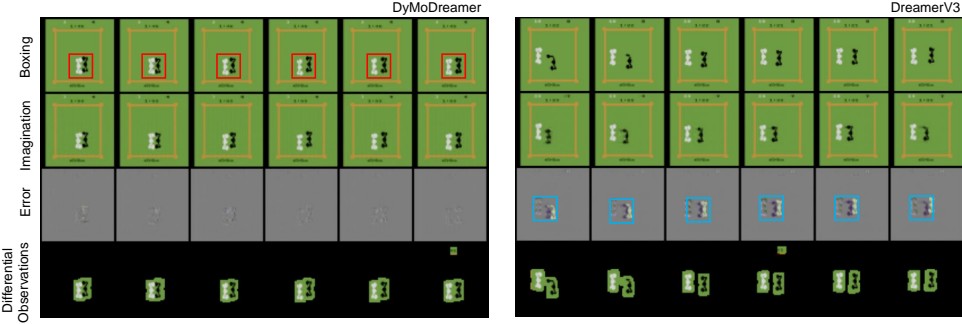

Figure 3: DyMoDreamer highlights the dynamic features. The players enclosed in red rectangles represent reward-relevant objects, which can be encoded into the dynamic modulation to enhance the RSSM. Blue rectangles mark the error regions where DreamerV3 exhibits hallucinations during the imagination reconstruction.

# 3 Experiments

## 3.1 Benchmarks and Results

We evaluate DyMoDreamer on several widely-used benchmarks for sample-effcient RL: Atari 100k, DeepMind Visual Control Suite and Crafter. Table 1 summarizes the aggregate scores of DyMoDreamer and baselines, with the associated performance detailed per-environment scores and curves provided in Appendix E.

**Atari 100k Benchmark**   We first evaluate DyMoDreamer on the Atari 100k benchmark, comprising 26 games from the Arcade Learning Environment, a widely adopted testbed for RL agents [15]. Agents are restricted to 100k actions (equivalent to 400k frames or $\sim 1.85$ hours of gameplay) for training before evaluation. This constraint emphasizes sample efficiency, a critical metric in reinforcement learning. DyMoDreamer significantly improves upon the base method's performance, and sets a new state-of-the-art benchmark with a mean human-normalized score (HNS) of 156.6%.

**DeepMind Visual Control Suite**   To validate DyMoDreamer's capacity for continuous control, we conduct experiments on the full set of 20 tasks in the DeepMind Visual Control Suite, challenging environments requiring policy learning from high-dimensional visual observations under a 1M step budget. In this suite, DyMoDreamer DyMoDreamer outperforms the newest and strongest existing baseline TWISTER [41], achieving a $5.5\%$ average performance improvement over the DreamerV3 baseline. The results indicate that the dynamic modulation can also yield performance gains for world models in tasks involving high-dimensional observations and continuous action spaces.

**Crafter Benchmark**   We finally complete the empirical validation using the Crafter [42] benchmark, which is a procedurally generated environment, inspired by the video game Minecraft, with visual inputs, a discrete action space and non-deterministic dynamics. In this benchmark, the background moves along with the agent, causing differential observations to capture the relative motion between the agent and the environmental components. Consequently, dynamic modulation enriches the information available for decision-making. On this benchmark, DyMoDreamer outperforms the IRIS series (the strongest transformer-based baselines)and achieves a $9.5\%$ performance gain over the DreamerV3 baseline.

## 3.2 Analysis and Implementation Details

We compared DyMoDreamer with methods leverage advanced architectures such as transformers and RSSMs, as well as diffusion models. Model-free and search-based approaches (e.g., BBF [43], EfficientZero [8]) are excluded, as our focus is the refinement of world models. Hyperparameters for DyMoDreamer follow DreamerV3 [14], except where explicitly noted, such as in the ablation studies.

Table 1: Aggregate scores of DyMoDreamer and baselines.

| | **Atari 100k** | | | |
|---|---|---|---|---|
| | OC-STORM (2025) | DIAMOND (2024) | DreamerV3 (2023) | DyMoDreamer (ours) |
| HNS Mean | 134.8% | 146% | 125% | **156.6%** |
| HNS Median | 43.8% | 37% | 49% | **71.3%** |
| | **DeepMind Visual Control Suite** | | | |
| | TD-MPC2 (2023) | TWISTER (2025) | DreamerV3 (2023) | DyMoDreamer (ours) |
| Task Mean | 720.9 | 801.8 | 786 | **832** |
| Task Median | 795.9 | **907.6** | 861 | 871 |
| | **Crafter** | | | |
| | IRIS (2023) | $\Delta$-IRIS (2024) | DreamerV3 (2023) | DyMoDreamer (ours) |
| Return @1M | 5.5 | 7.7 | 9.4 | **10.3** |

DyMoDreamer demonstrates superior performance compared to previous methods in environments where the key objects related to rewards are sparse and independent with each other, such as *Pong* and *Finger Spin*. This advantage can be attributed to the accurate capturing of dynamic modulators, which explicitly enhances the world model awareness of temporal features in the observations. DyMoDreamer also excels in tasks with distinct phases, such as *Krull*, and Crafter, as the predictable dynamic modulators explicitly guides the world model to capture transitions between task stages. Moreover, even when the reward signal is not fully captured by the dynamic information, DyMoDreamer still yields performance gains by explicitly retaining the stochastic representations, for example, *Gopher*. It's noteworthy that this enhancement is achieved without leveraging any additional techniques from the computer vision domain, thereby incurring no extra oversized computational overhead. By introducing dynamic modulation into the world model, we fully exploit the intrinsic potential of the RSSM framework, thereby enhancing the significance of our contribution. In our experiments, we use a machine with an NVIDIA 4090 graphics card with 8 CPU cores and 24 GB RAM. Training DyMoDreamer on one Atari game for 100k steps took roughly 5.5 hours in our JAX implementation.

## 4    Ablation Studies

Our experiments have shown incorporating dynamic modulators into the world model can greatly enhance the final outcomes. To further investigate this, we perform ablation studies on the two novel components introduced in DyMoDreamer: the dynamic modulation (Section 4.1) and the differential divergence regularization $\mathcal{L}_{\text{reg}}$ (Section 4.2). We select four representative game environments: *Boxing*, featuring large and sparse dynamic objects; *Krull*, characterized by small and numerous dynamic objects; *Pong*, with small and sparse dynamic objects; and *Road Runner*, which includes both small and large dynamic objects.

### 4.1    Removing Dynamic Modulation

We first perform an ablation study where the dynamic modulators are retained while its integration into RSSM is removed, in order to assess the necessity of incorporating dynamic modulation for achieving performance gains. In this setting, the sequence model in Equation 3 degenerates to $h_t = f_\phi(h_{t-1}, z_{t-1}, a_{t-1})$, where the dynamic modulation is only involved in encoding and image reconstruction. Additionally, the constraints on the modulators $d_t$ are removed from both $\mathcal{L}_{\text{dyn}}$ and $\mathcal{L}_{\text{dyn}}$. As shown in Figure 4, removing the dynamic modulation mechanism causes a notable performance drop. Simply appending differential observations, without integrating them into the RSSM, fails to direct the world model's focus toward dynamic patterns. This result is comparable to the ablation in Appendix J.1, where increasing the dimensionality of $z_t$ alone offers limited benefits. These findings indicate that the dynamic modulation should be embedded into the sequence model to

be effective. By integrating information from the predictable modulators into the recurrent state, the agent gains access to richer environment cues during decision-making, beyond merely increasing the dimensionality of the VAE.

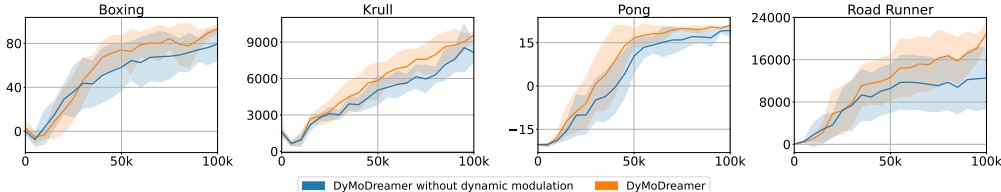

Figure 4: Ablation study on the dynamic modulation.

## 4.2 Differential Divergence Regularization

The differential divergence regularization $\mathcal{L}_{\mathrm{reg}}$ addresses the limitations of mean-squared error (MSE) loss, which primarily focuses on intra-frame differences. Therefore, we remove this component from DyMoDreamer and observe a performance drop (Figure 5), which demonstrates that the differential divergence regularization acts as a complementary supervisory signal that works in concert with reconstruction objectives, and contributes improvement through its unique ability to constrain the temporal consistency in the reconstruction process. Removing the differential divergence regularization leads to a performance drop; however, thanks to the dynamic modulation mechanism, the model still surpasses vanilla DreamerV3.

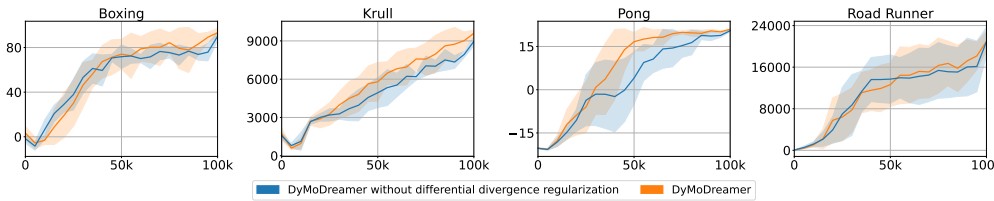

Figure 5: Ablation study on the differential divergence regularization.

## 5 Related Work

World model-based reinforcement learning enables agents to learn via imagination, improving sample efficiency by predicting future states and rewards from compact latent representations. Early works like SimPLe [12] and Dreamer [11] pioneered latent dynamics modeling for decision-making. DreamerV2 [13] introduced discrete latents to mitigate compounding errors, and DreamerV3 [14] extended this to diverse domains with fixed hyperparameters. HarmonyDream [44] further refined training via adaptive loss balancing. Recent advances leverage transformers (e.g., STORM [25] and OC-STORM [21]) for scalable sequence modeling, with innovations like object-centric prediction. The IRIS series [9, 22] advanced tokenized imagination by employing VQ-VAE [45] to encode observations into discrete codes, compressing visual inputs into a compact symbolic space that facilitates long-horizon predictions. Meanwhile, DIAMOND [24] applies score-based diffusion models in pixel space for detailed visual modeling. Hierarchical world models (e.g., THICK [46], HIEROS [47], and Puppeteer [48]) further improve temporal abstraction, enabling long-horizon reasoning and coordination in complex environments.

## 6 Conclusion

In this work, we propose DyMoDreamer, a novel world modeling approach with dynamic modulation, which outperforms previous model-based methods in Atari 100k, DeepMind Visual Control and Crafter benchmarks, setting a new record for the RSSM architecture [17]. DyMoDreamer integrates dynamic modulation into RSSM, enabling the agent to effectively capture decision-relevant dynamic features and temporal information within the environment throughout the decision-making process.

We capture dynamic features and temporal information by encoding differential observations, and enhance dynamic modulation with specialized dynamics and representation losses to improve its ability to integrate dynamic information. This design significantly improves the performance of the world model. Our work demonstrates that the RSSM framework remains highly promising and opens numerous avenues for future research. One potential direction is to introduce soft predictive constraints on future states during end-to-end training could prove beneficial [49, 50]. We believe that designing more human-like world model algorithms by learning from human's cognitive behaviors holds significant potential for impact and represents an exciting path for exploration.

## 7 Acknowledgments and Disclosure of Funding

We would like to thank all anonymous reviewers for their time, constructive comments, and engaging discussions. This work was supported by the National Key Research and Development Program of China under Grant No. 2022ZD0119302, and the National Natural Science Foundation of China under Grants U23B2059, 62173034, and 62495090.

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

# A  Further Related Work

Model-based RL utilizing world models enables "learning through imagination" to provide greater data efficiency and offers a promising approach [12] . These algorithms use self-supervised manners to construct parameterized simulation world models of the environments, leveraging losses such as reconstructing observations with decoders, predicting rewards and states at the next step, and the consistency between prior and posterior of latent states. Agent learns behaviors by the imagination trajectories based on world model predictions, reducing the reliance on environment interactions and improving the sample efficiency. SimPLe [12] applied world models to enable agents to focus on sample efficiency, and solve Atari games with fewer interactions than model-free Rainbow [51]. Dreamer [11] pioneered the reinforcement learning directly from the latent space of a recurrent state space model. Building on this, DreamerV2 [13] demonstrated that employing discrete latent variables mitigates compounding errors, while DreamerV3 [14] further advanced the approach, achieving human-level performance across diverse domains using fixed hyperparameters. HarmonyDream [44] employed a harmonious loss to dynamically adjust the weights of different loss terms during training. This approach ensures that the losses of various tasks, such as image reconstruction and reward modeling, remain within the same order of magnitude throughout the end-to-end training process.

Following the success of the Dreamer algorithm [11], the transformer architectures [18] are also rapidly adopted for its superior training efficiency and favorable scaling properties. TWM [52] utilized the Transformer-XL [20] architecture to learn long-term dependencies while staying computationally efficient. IRIS [9] and $\Delta-$IRIS [22] utilized VQ-VAE [45] to build a language of image tokens to represent observations and make decisions based on the reconstructed observations. Building upon the IRIS framework, REM introduced a novel parallel observation prediction mechanism to enhance the imagination, and achieves performance surpassing the baseline [53]. Simulus [54] proposes a highly modular token-based world model that integrates a multi-modality tokenization framework, intrinsic motivation, prioritized replay, and a regression-as-classification strategy for predicting rewards and returns. STORM [25] used GPT-like architecture with a different tokenization approach, which captures long-term dependencies through attention mechanism and achieves impressive performance on the Atari 100k benchmark. OC-STORM extracted object features through a parameter-frozen pre-trained foundation vision model "Cutie" [55] to apply both these object features and the raw observations as inputs for training an object-centric world model, which predicts the dynamic objects in the environment while considering the relationships between different objects and the background. Additionally, DIAMOND [24] introduced a diffusion-based world model that operates entirely within the image space, enabling more accurate modeling of critical visual details in the world model.

Another type of world model adopts a hierarchical structure, this architecture operates at different time scales, allowing the agent to learn and make decisions across multiple levels of abstraction. [46] deploys a hierarchy of world models with a look ahead trajectory search approach called THICK, which learns the world model via discrete latent dynamics. The lower level of THICK updates parts of its latent state sparsely over time, forming invariant contexts. The higher level exclusively predicts situations that involve context changes. HIEROS proposed an S5 layer-based world model, which learned time abstracted world representations and imagines trajectories at multiple time scales in latent space [47]. Puppeteer [48] proposed hierarchical world model in which a high-level agent generates commands based on visual observations for a low-level agent to execute, and produces highly performant control policies in 8 tasks with a simulated 56-DoF humanoid [56], while synthesizing motions that are broadly preferred by humans.

# B  Limitations

DyMoDreamer improves the temporal world modeling through dynamic modulation, yet fundamental limitations of MBRL remain. Accurately capturing environment dynamics, especially in visually complex or highly uncertain environments continues to be a key challenge. Compounding errors during long-horizon imagination hinder policy performance, and recurrent latent models offer limited capacity for long-term memory. Although DyMoDreamer partially addresses these issues via architectural enhancements, future work may benefit from integrating memory-augmented transformers or hierarchical abstractions to improve long-term reasoning. Moreover, despite gains in sample efficiency, the trade-off between model fidelity and policy stability persists. Generalization to real-world domains and robustness under distributional shifts remain critical open problems.

## C  Broader Impacts

Training agents for real-world applications, such as robotics and autonomous driving, is challenging due to high costs, safety risk, and potential harm to humans. To mitigate these concerns, we conduct all experiments in simulated environments, eliminating risks associated with real-world training. We introduce DyMoDreamer, a world model with dynamic modulation that enables agents to learn through imagination rather than direct physical interactions. By reducing reliance on real-world data collection, our approach enhances the safety and feasibility of deploying model-based reinforcement learning in safety-critical domains. This work contributes to the development of efficient and ethically responsible autonomous systems, addressing key concerns in real-world AI deployment.

## D  Policy Learning

The policy learning approach closely follows that of DreamerV3 [14], with modifications specific to our method. To our DyMoDreamer, the actor and critic operate on the concentration states $s_t = \{z_t, h_t, d_t\}$. We select actions by sampling from the actor network and aim to maximize the return $R_t = \sum_{\tau=1}^{\infty} \gamma^\tau r_{t+\tau}$ with a discount factor $\gamma = 0.997$ for each state. The critic learns [57] to approximate the distribution of returns for each state under the current actor behavior:

$$
\begin{aligned}
\text{Actor:} \quad & a_t \sim \pi_\theta(a_t \,|\, s_t) \\
\text{Critic:} \quad & V_\psi(R_t \,|\, s_t) \approx \mathbb{E}_{\pi_\theta, p_\phi}\Big[ \sum_{\tau=0}^{\infty} \gamma^\tau r_{t+\tau} \Big],
\end{aligned}
\tag{11}
$$

where the predicted values from the critic is a expectation of the distribution it predicts. We adopt the actor and critic learning settings from DreamerV3. The complete loss of the actor-critic algorithm is described by Equation (7), where $\hat{r}_t$ corresponds to the reward predicted by the world model, and $\hat{c}_t$ represents the predicted continuation flag. The critic learns to approximate the distribution of the return estimates $R_t^\lambda$ using the maximum likelihood loss:

$$
\mathcal{L}(\psi) = \frac{1}{BL} \sum_{n=1}^{B} \sum_{t=1}^{L} \Big[ \big( V_\psi(s_t) - \text{sg}(R_t^\lambda) \big)^2 + \big( V_\psi(s_t) - \text{sg}(V_{\psi^{\text{EMA}}}(s_t)) \big)^2 \Big],
\tag{12}
$$

where the constant $L$ represent the imagination horizon and $V_{\psi^{\text{EMA}}}$ is the exponential moving average (EMA) of the critic network $\psi$ for stabilizing training and preventing overfitting [58].

For the actor network, we use the Reinforce estimator for both discrete and continuous actions, resulting in the surrogate loss function:

$$
\mathcal{L}(\theta) = \frac{1}{BL} \sum_{n=1}^{B} \sum_{t=1}^{L} \Big[ -\text{sg}\Big( \frac{R_t^\lambda - V_\psi(s_t)}{\max(1, S)} \Big) \ln \pi_\theta(a_t | \mathbf{s}_t) - \eta H(\pi_\theta(a_t | s_t)) \Big],
\tag{13}
$$

where $H(\cdot)$ denotes the entropy of the policy distribution, while constants $\eta$ represent the coefficient for entropy loss. The $\lambda$-return $R_t^\lambda$ [59] is recursively defined as follows

$$
\begin{aligned}
R_t^\lambda &= r_t + \lambda c_t \Big[ (1 - \lambda) V_\psi(s_{t+1}) + \lambda R_{t+1}^\lambda \Big], \\
R_L^\lambda &= V_\psi(s_L).
\end{aligned}
\tag{14}
$$

The normalization ratio $S$ utilized in the actor loss (14) is defined in (15), which is computed as the range between the $5^{\text{th}}$ and $95^{\text{th}}$ percentiles of the $\lambda$-return $R_t^\lambda$ across the batch:

$$
S = \text{percentile}(R_t^\lambda, 95) - \text{percentile}(R_t^\lambda, 5).
\tag{15}
$$

## E  Experimental Details

### E.1  Atari 100k Experiments

We evaluate agent performance over 100 episodes using the final model checkpoint, saved every $2,500$ steps. The reward function in *Freeway* is sparse since the agent is only rewarded when

it completely crosses the road. This poses an exploration problem for newly initialized agents because a random policy will almost surely never obtain a non-zero reward with a 100k frames budget. Inspired by the trick in IRIS [9], we opted for the simpler strategy of having a fixed epsilon-greedy parameter and using the policy. However, we reduced the sampling temperature from 1 to 0.0001 in *Freeway*, in order to avoid random walks that would not be conducive to learning in the early stages of training. The human-normalized score is calculated as $\tau = \frac{A-R}{H-R}$, where $A$ is the agent's score, $R$ is the score of a random policy, and $H$ represents the average human score [5]. To determine $H$, the human players undergo familiarization with the game under identical constraints. We compare DyMoDreamer with state-of-the-art model-based RL methods, including OC-STORM [21], DreamerV3 [14], HarmonyDream [44], and DIAMOND [24]. The task scores are shown in Table 2 and the curves are shown in Figure 6.

Table 2: Game scores and overall human-normalized performance on the 26 games in the Atari 100k benchmark. Following the conventions of [14], scores within $5\%$ of the best are highlighted in bold.

| Game | Random | Human | STORM (2023) | OC-STORM (2025) | DIAMOND (2024) | DreamerV3 (2023) | Harmony DreamerV3 | DyMoDreamer (ours) |
|---|---|---|---|---|---|---|---|---|
| Alien | 228 | 7128 | 984 | **1101** | 744 | **1118** | 890 | 971 |
| Amidar | 6 | 1720 | 205 | 163 | **226** | 97 | 141 | 154.2 |
| Assault | 222 | 742 | 801 | 1270 | **1526** | 683 | 1003 | 902.1 |
| Asterix | 210 | 8503 | 1028 | 1754 | **3698** | 1062 | 1140 | 1195.5 |
| Bank Heist | 14 | 753 | 641 | 1075 | 20 | 398 | 1069 | **1222.2** |
| Battle Zone | 2360 | 37188 | 13540 | 4590 | 4702 | **20300** | 16456 | 16240 |
| Boxing | 0 | 12 | 80 | **92** | 87 | 82 | 80 | **93.6** |
| Breakout | 2 | 30 | 16 | 53 | **132** | 10 | 53 | 40.2 |
| Chopper Command | 811 | 7388 | 1888 | **2090** | 1370 | 2222 | 1510 | 740.4 |
| Crazy Climber | 10780 | 35829 | 66776 | 84111 | **99168** | 86225 | 82739 | 82569.4 |
| Demon Attack | 152 | 1971 | 165 | 411 | 288 | **577** | 203 | 248.3 |
| Freeway | 0 | 30 | 0 | 0 | **33** | 0 | 0 | 6.4 |
| Frostbite | 65 | 4335 | 1316 | 260 | 274 | **3377** | 679 | 722 |
| Gopher | 258 | 2413 | 8240 | 4457 | 5898 | 2160 | 13043 | **13456.9** |
| Hero | 1027 | 30826 | 11044 | 6441 | 5622 | **13354** | **13378** | 9874.1 |
| James Bond | 29 | 303 | 509 | 347 | 427 | **540** | 317 | **541.8** |
| Kangaroo | 52 | 3035 | 4208 | 4218 | **5382** | 2643 | **5118** | 4788.8 |
| Krull | 1598 | 2666 | 8413 | **9715** | 8610 | 8171 | 7754 | **9624.8** |
| Kung Fu Master | 256 | 22736 | 26182 | 24988 | 18714 | **25900** | 22274 | 25641.4 |
| Ms Pacman | 307 | 6952 | **2673** | 2401 | 1958 | 1521 | 1681 | 1838.6 |
| Pong | -21 | 15 | 11 | **20.6** | **20** | -4 | 19 | **20.9** |
| Private Eye | 25 | 69571 | **7781** | 85 | 114 | 3238 | 2932 | 3041.7 |
| Qbert | 164 | 13455 | **4522** | **4546** | **4499** | 2921 | 3933 | 1736.1 |
| Road Runner | 12 | 7845 | 17564 | **20482** | 20673 | 19230 | 14646 | **20971.8** |
| Seaquest | 68 | 42055 | 525 | 712 | 551 | **962** | 665 | 591.8 |
| Up N Down | 533 | 11693 | 7985 | 6623 | 3856 | **46910** | 10874 | 22321.5 |
| Human Mean | 0% | 100% | 126.7% | 134.8% | 146% | 125% | 136.5% | **156.6%** |
| Human Median | 0% | 100% | 58.4% | 43.8% | 37% | 49% | 67.1% | 71.3% |

## E.2 DeepMind Visual and Proprio Control Suite Experiments

The DeepMind Visual Control Suite is a widely used benchmark for visual locomotion. We train and evaluate our DyMoDreamer on all 20 continuous control tasks where the agent receives only high-dimensional images as observations and has a budget of 1M environment steps. We evaluate the agent performance by conducting 100 evaluation episodes for the final checkpoint and obtained the average score. DyMoDreamer is compared with DreamerV3, TD-MPC2 [3], and TWISTER [41], which are recent model-based apporaches applied to continuous control. The task scores are shown in Table 3 and the curves are shown in Figure 7.

The results demonstrate the effectiveness of DyMoDreamer on DMC visual tasks. Our method greatly unleash the potential of RSSM. Similar to the results in Atari 100k, DyMoDreamer benefits from the dynamic modulation and shows superior performance compared to DreamerV3 in environments where key objects are sparse and independent of each other.

To evaluate the applicability of DyMoDreamer in state-based environments, we conducted experiments on DeepMind Proprio, which is a proprioceptive control benchmark where observations consist of structured state vectors such as velocity and position. In this setting, observation differencing naturally reduces to state differencing, with temporal dynamics being more explicitly encoded. Consequently, differential modulation provides similar benefits in non-visual settings, since such

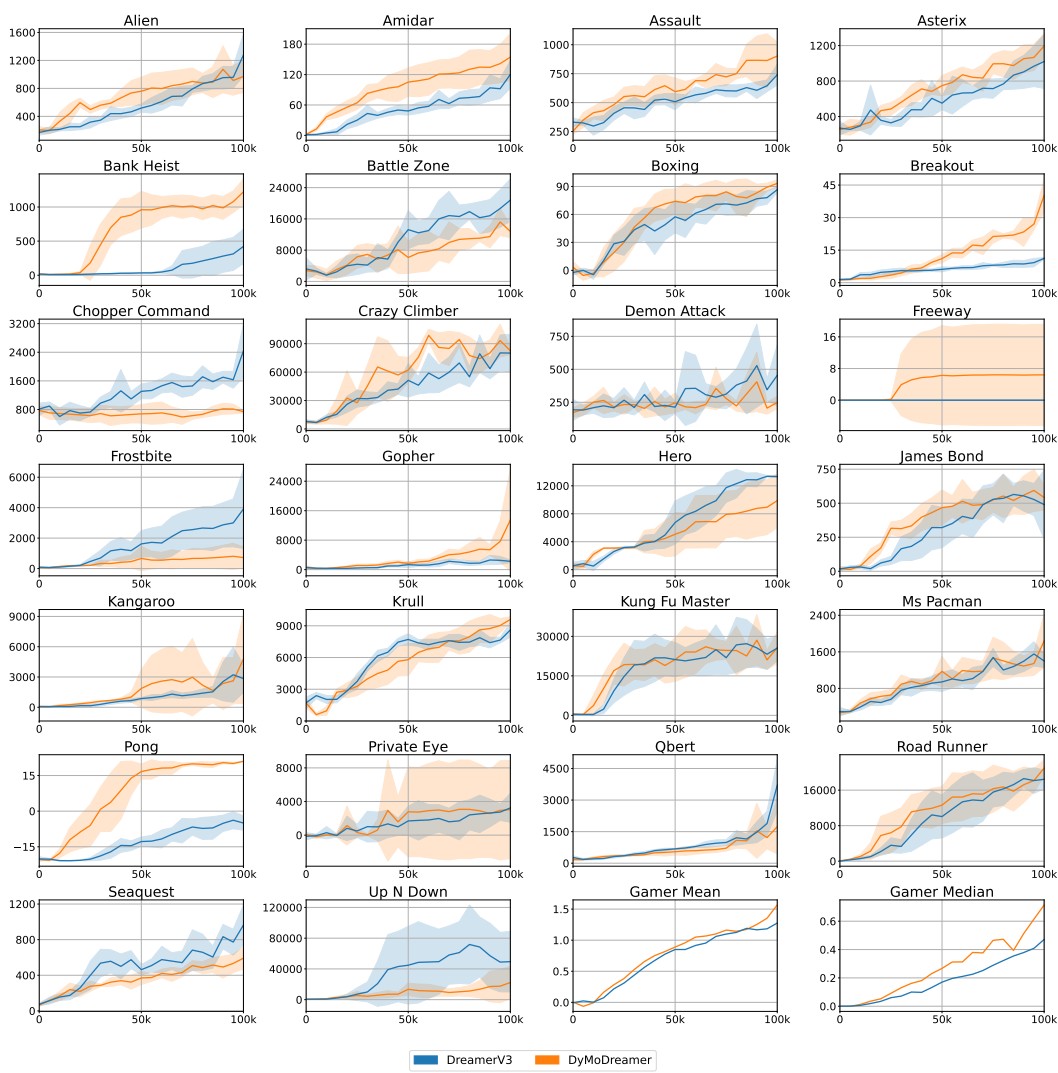

Figure 6: Evaluation curves of DyMoDreamer on the Atari 100k benchmark for individual games (400K environment steps). The solid lines represent the average scores over 5 seeds, and the filled areas indicate the standard deviation across these 5 seeds.

differences also carry meaningful physical semantics similar to the differential observations—for instance, differential positions implicitly encode velocity, while the differential velocities reflect acceleration.

### E.3 Crafter Benchmark Experiments

In the Crafter [42] benchmark, the agent's goal is to solve as many tasks as possible, e.g. slaying mobs, crafting items, and managing health indicators during each episode. Regarding baselines, we compare DyMoDreamer with IRIS, $\Delta$-IRIS and DreamerV3, and evaluate each run by computing the average return over 100 test episodes every 1M frames, the curve is shown in Figure 8.

### E.4 Deepmind Lab Benchmark Experiments

The Deepmind lab [42] benchmark features 3D environments that test spatial and temporal reasoning. To the concerns about the evaluation on more explicitly egocentric environments, we conducted additional experiments on the DeepMind Lab benchmark to further validate DyMoDreamer's effectiveness under rapid viewpoint changes. We focus on several representative tasks with relatively dense

Table 3: Task scores and mean scores of baselines and our DyMoDreamer

| Task | TD-MPC2 (2023) | TWISTER (2025) | DreamerV3 (2023) | DyMoDreamer (ours) |
|---|---|---|---|---|
| Environment steps | 1M | 1M | 1M | 1M |
| Acrobot Swingup | 216 | 239 | 229 | **447** |
| Ball In Cup Catch | 717 | 967 | **972** | 967 |
| Cartpole Balance | 931 | 998 | 993 | **999** |
| Cartpole Balance Sparse | **1000** | **1000** | 964 | 999 |
| Cartpole Swingup | 808 | 819 | 861 | **864** |
| Cartpole Swingup Sparse | 739 | 735 | 759 | **765** |
| Cheetah Run | 550 | 694 | 836 | **868.7** |
| Finger Spin | **986** | 976 | 589 | 968 |
| Finger Turn Easy | 789 | **924** | 878 | 906 |
| Finger Turn Hard | 872 | **910** | 904 | 873 |
| Hopper Hop | 211 | **314** | 227 | 290 |
| Hopper Stand | 803 | 932 | 903 | **943** |
| Pendulum Swingup | 743 | **832** | 744 | 808.9 |
| Quadruped Run | 362 | 652 | 617 | **770.4** |
| Quadruped Walk | 253 | **905** | 811 | 792.0 |
| Reacher Easy | **971** | 933 | 951 | 896.8 |
| Reacher Hard | **877** | 566 | **862** | 821.9 |
| Walker Run | 728 | 711 | 684 | **776.3** |
| Walker Stand | 916 | **977** | 976 | 929.6 |
| Walker Walk | 945 | 951 | 961 | **962.0** |
| Task mean | 721 | 802 | 786 | **832** |
| Task median | 796 | **908** | 861 | 871 |

Table 4: Game scores in the DeepMind Proprio benchmark.

| | DreamerV3 | DyMoDreamer |
|---|---|---|
| Acrobot Swingup | 134 | **225** |
| Cheetah Run | 614 | **625** |
| Swingup | **931** | **932** |

reward signals, which are more likely to exhibit performance improvements within 100M steps to assess DyMoDreamer's performance. The scores are shown in Table 5. DyMoDreamer outperforms the baseline on 4 out of 5 tasks, demonstrating competitive performance particularly on tasks with localized motion rewards (e.g., Explore Goal Locations Small, Explore Object Rewards Many) and dynamic visual input (e.g., Rooms Watermaze). However, in Rooms Keys Doors Puzzle, which emphasizes long-horizon memory and discrete planning over motion cues, DreamerV3 shows a slight advantage, consistent with our analysis of task suitability. In such settings, although the differential observations may not isolate only the task-relevant objects, they nonetheless effectively suppress large portions of decision-irrelevant background. This facilitates more efficient exploration by directing the agent's attention toward reward-associated regions of the scene. Furthermore, differential observations are not merely a visual processing technique, they also serve as an embedding of temporal information into the model's representation, enriching its capacity to reason over dynamic sequences. These results suggest that DyMoDreamer's dynamic modulation mechanism offers generalizable benefits in visually rich and moderately egocentric environments, though it may be less effective in domains requiring complex, abstract reasoning or long-term symbolic memory.

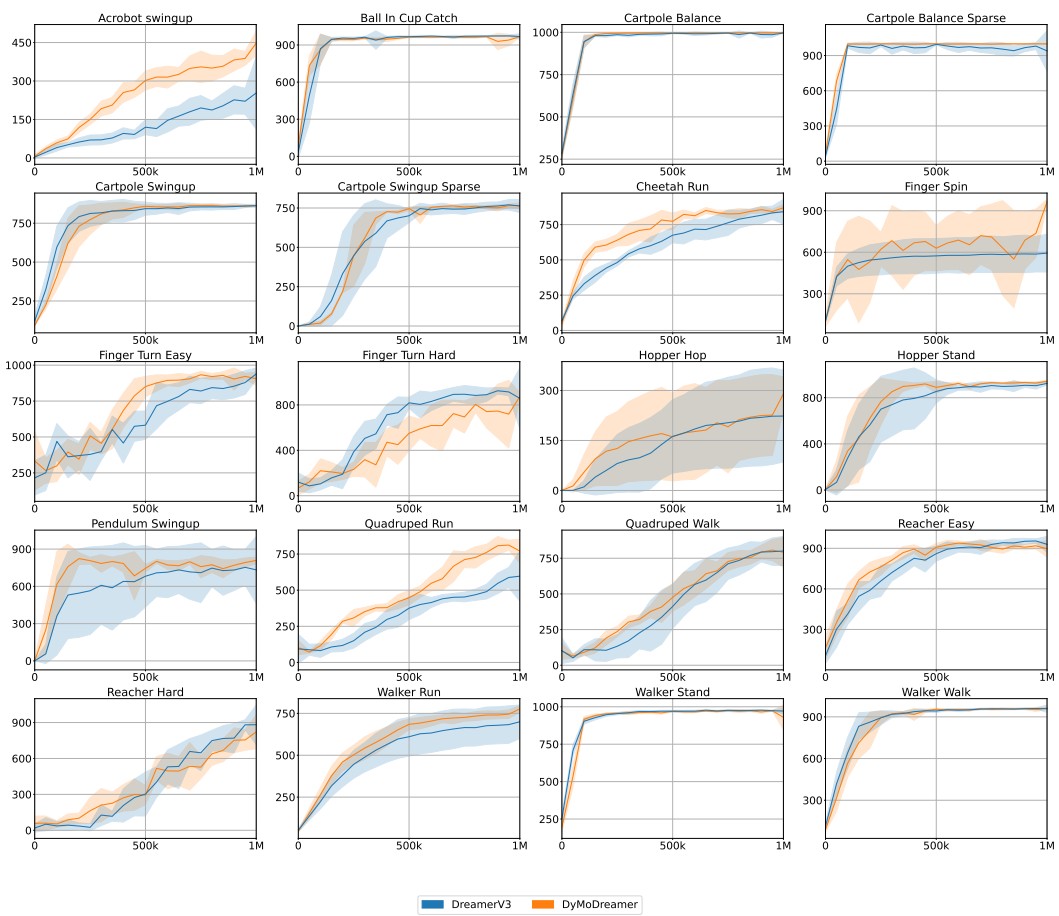

Figure 7: Evaluation curves of DyMoDreamer on the DMC suite. The solid lines represent the average scores over 5 seeds, and the filled areas indicate the standard deviation across these 5 seeds.

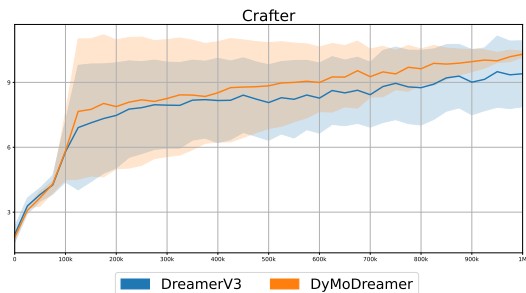

Figure 8: Evaluation curves of DyMoDreamer on the Crafter benchmark. The solid lines represent the average scores over 5 seeds, and the filled areas indicate the standard deviation across these 5 seeds.

Table 5: Task scores in the Deepmind Lab benchmark.

|  | DreamerV3 | DyMoDreamer |
|---|---|---|
| Explore Goal Locations Small | 372.8 | **382.1** |
| Explore Object Rewards Many | 58.1 | **60.0** |
| Lasertag Three Opponents Small | 18.8 | **18.9** |
| Rooms Keys Doors Puzzle | **42.4** | 41.2 |
| Rooms Watermaze | 26.1 | **27.2** |

# F    Differential Observations

Figure 9, 10 and 11 illustrate the differential observations processed using differential mask constructed by Section 2.1. Compared to directly employing frame-wise differences, the masks generated through dilated convolutions capture dynamic objects more comprehensively, ensuring a more complete representation of motion-related features. Moreover, this demonstrates that the dynamic modulation is effectively encoded from the decision-related dynamic objects within the observations.

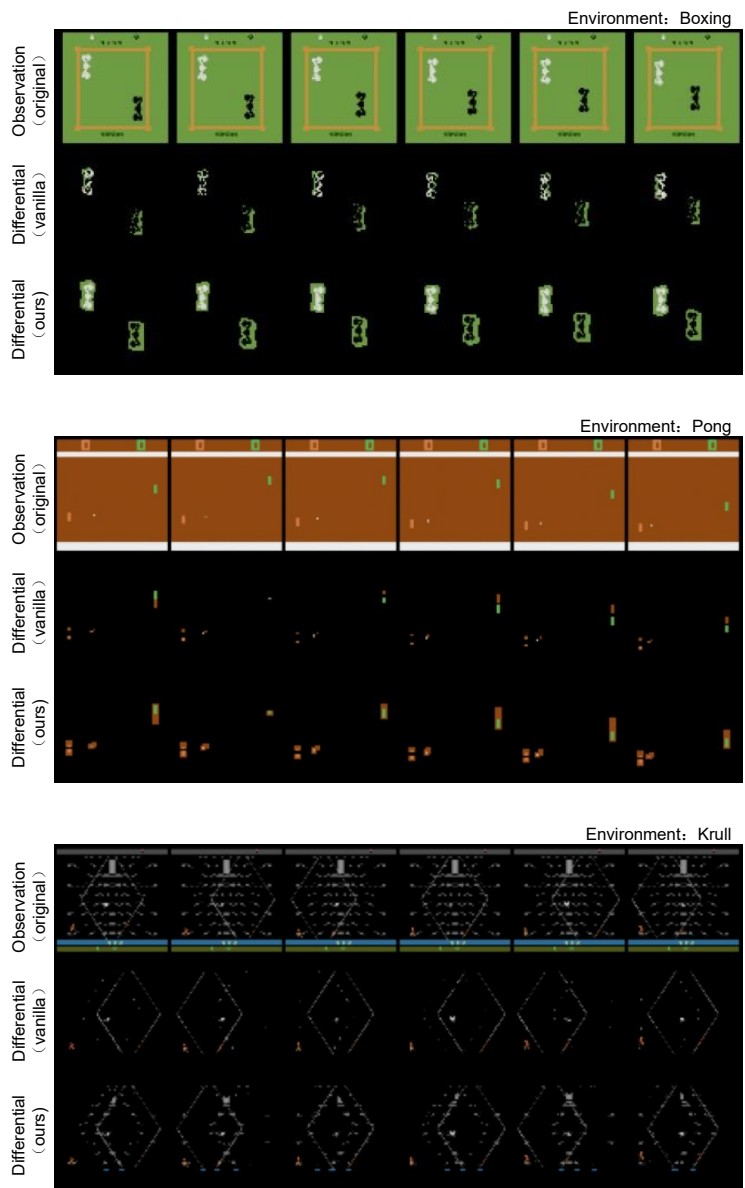

Figure 9: Differential observations for *Boxing* (the first subfigure), *Pong* (the second subfigure), and *Krull* tasks (the last subfigure). Each subfigure comprises three rows, and each illustrates a distinct form of observation. The first row displays the raw observations, the second row shows observations processed using vanilla frame differencing (2), and the third row presents the differential observations constructed using our proposed approach, which combines frame differencing with dilated convolutions.

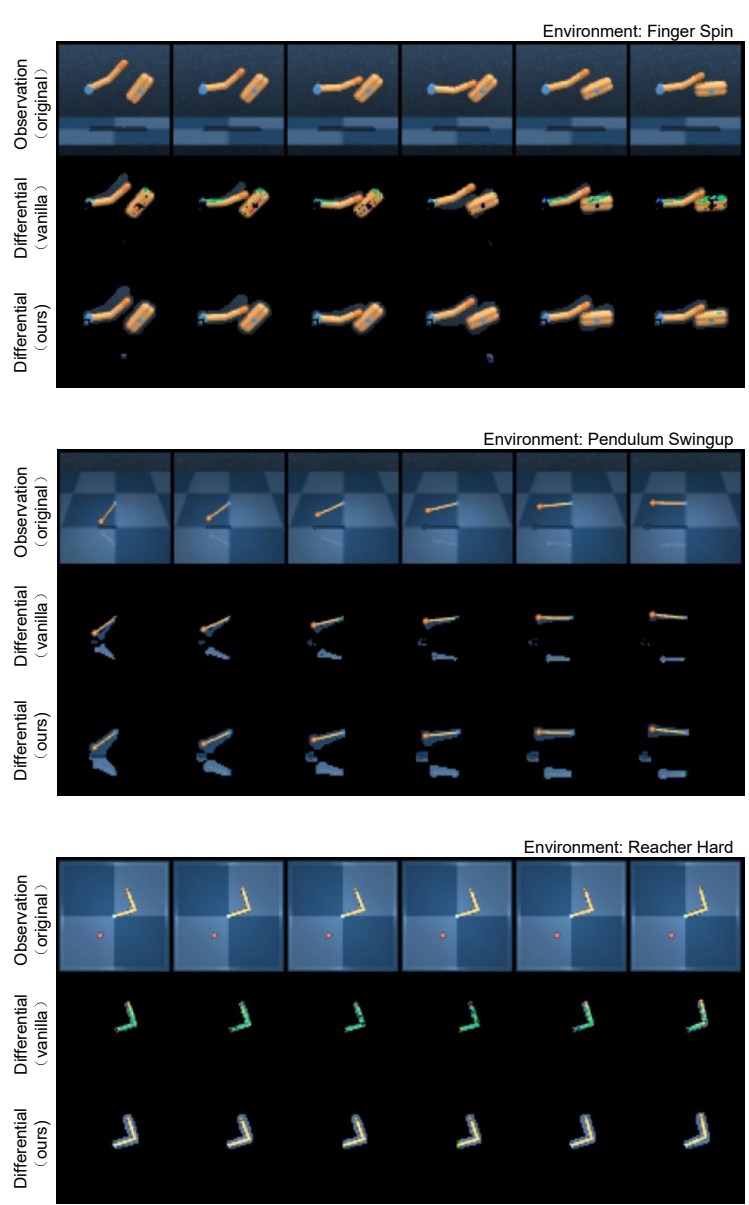

Figure 10: Differential observations for *Finger Spin* (the first subfigure), *Pendulum Swingup* (the second subfigure), and *Reacher Hard* (the last subfigure) tasks. Each subfigure comprises three rows, and each illustrates a distinct form of observation. The first row displays the raw observations, the second row shows observations processed using vanilla frame differencing (2), and the third row presents the differential observations constructed using our proposed approach, which combines frame differencing with dilated convolutions.

## G    Alternative Differencing Strategies

A naive separation of dynamic and static components may introduce inductive biases in certain environments. To mitigate such issues (dynamic backgrounds or environmental noise), DyMoDreamer is designed to be highly flexible and accommodates a variety of differencing strategies, enabling seamless adaptation to diverse visual environments. While our default implementation adopts frame-wise pixel differencing to highlight dynamic cues, the architecture readily supports alternative formulations such as moving average difference, temporal skip differencing (Appendix J.6), or

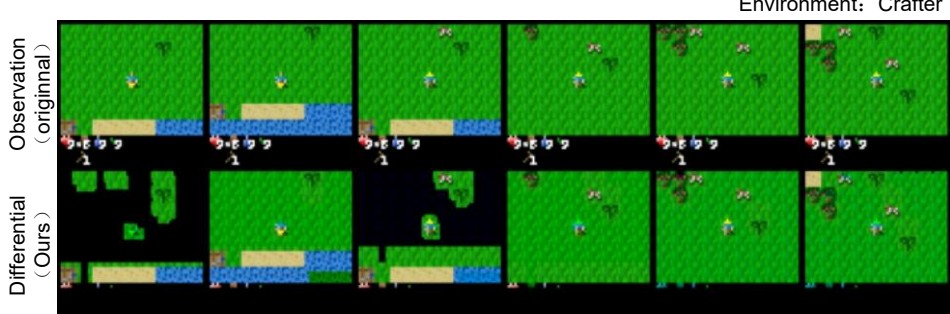

Figure 11: Differential observations for Crafter. The first row displays the raw observations, the second row presents the differential observations constructed using our proposed approach, which combines frame differencing with dilated convolutions.

multi-frames differencing. Importantly, frame differencing has long been recognized in computer vision as a lightweight yet effective method for motion detection, foreground segmentation, and event-based visual processing, due to its ability to isolate dynamic elements from static backgrounds [60]. By leveraging this principle within the context of world model, DyMoDreamer inherits its computational efficiency and interpretability, while maintaining the temporal information with the dynamic modulation mechanism. DyMoDreamer demonstrates that the differencing-based dynamic modulation not only enriches temporal representation but also offers a flexible interface to incorporate various differencing strategies, making it adaptable to a broad range of task scenarios.

**Moving Average Difference**   We formally express the replacement of frame differencing (Equation 2) with moving average differencing as follows:

$$D(o_{i,h,w,c}) = \begin{cases} 1, & \text{if } \|o_{i,h,w,c} - \text{avg}_i\|_2 > \epsilon \\ 0, & \text{others} \end{cases} \tag{16}$$

where $\text{avg}_i$ is defined as:

$$\text{avg}_i = \begin{cases} \dfrac{1}{l} \displaystyle\sum_{k=i-l}^{i-1} o_{k,h,w,c} & \text{if } i \geq l \\ \dfrac{1}{i} \displaystyle\sum_{k=0}^{i-1} o_{k,h,w,c}, & i \leq l \end{cases} \tag{17}$$

where $l$ is the temporal averaging window size. This approach demonstrates superior performance over naive frame differencing in egocentric vision tasks, as evidenced in domains like *Battlezone* (Table 6).

**Multi-frame Logical Differencing**   We further propose a novel multi-frame logical-relational differencing that stabilizes the differencing output $D(o_{i,h,w,c})$ in Equation 2 by applying logical AND operations (that is, only pixels whose masks were 1 in the past few times will be displayed as 1 in the final matrix.) across temporally displaced frames:

$$D_{\log}(o_{i,h,w,c}) = \bigwedge_{k \in \Omega(t)} D_k, \tag{18}$$

where $\Omega(t) = \{i - \Delta, i\}$ and $\Delta$ is the symmetric temporal window size. This method demonstrates superior performance in environments with higher temporal continuity, as evidenced in domains like (Table 6).

Table 6: DyMoDreamer with different differencing strategies.

| | **DyMoDreamer** | | |
| --- | --- | --- | --- |
| | Moving Average Differencing | Multi-frame Logical Differencing | Vanilla |
| Battle Zone | **20351** | - | 16240 |
| Road Runner | - | **21378** | 20972 |

## H  Static Factors

We extend DyMoDreamer by introducing a dedicated decoder for its stochastic representations $z_t$, revealing an emergent specialization in the learned latent space:

$$
\begin{aligned}
\text{Stochastic encoder:} \quad & z_t \sim q_\phi\big(z_t \mid h_t, o_t\big) \\
\text{Dynamic encoder:} \quad & d_t \sim q_\phi\big(d_t \mid h_t, o'_t\big) \\
\text{Decoder:} \quad & \hat{\text{dyn}}_t = p_\phi\big(\hat{\text{dyn}}_t \mid h_t, z_t, d_t\big) \\
\text{Static decoder:} \quad & \hat{\text{sta}}_t = p_\phi\big(\hat{\text{sta}}_t \mid h_t, z_t\big).
\end{aligned}
\tag{19}
$$

The reconstructed observations $\hat{o}_t$ is calculated by:

$$
\hat{o}_t = \hat{\text{dyn}}_t + \hat{\text{sta}}_t.
\tag{20}
$$

Critically, we maintain the original end-to-end loss 5 without modification, and validate the representational specialization through natural reconstructions, which is shown in Figure 12. The results demonstrate a clear representational dissociation: stochastic representations encoding predominantly capture static environmental factors and passive dynamics (e.g., static backgrounds and the black NPC in *Boxing*), while the dynamic modulation features specialize in agent-controllable elements (e.g., the controlled white player in *Boxing*). This emergent specialization suggests an natural segregation within the model's representational architecture like the denoised Markov decision process [61], since we do not explicitly assign reconstruction targets to different decoders.

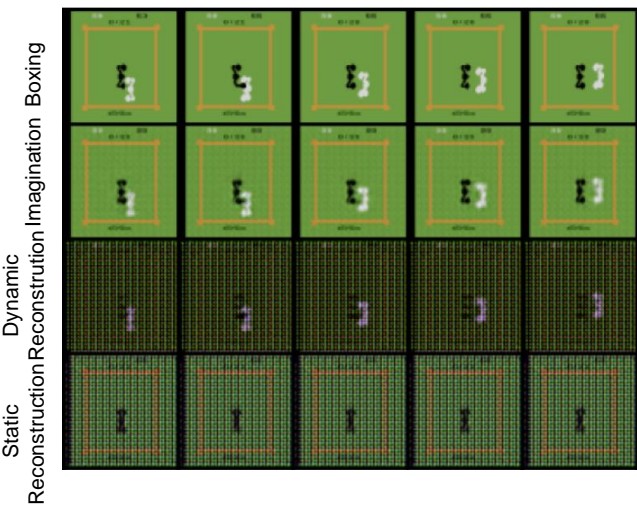

Figure 12: The first row shows ground-truth trajectories under the same action sequence. The second row presents imagined trajectories reconstructed from the predicted priors. The third row shows dynamic decoding results based on dynamically modulated features $d_t$. The fourth row shows static decoding results based on stochastic representations $z_t$.

# I Generality

To further validate the effectiveness of our dynamic modulation mechanism, we integrated it into STORM [25], a recent efficient transformer-based world model. As shown in the results below, this integration also leads to clear performance gain.

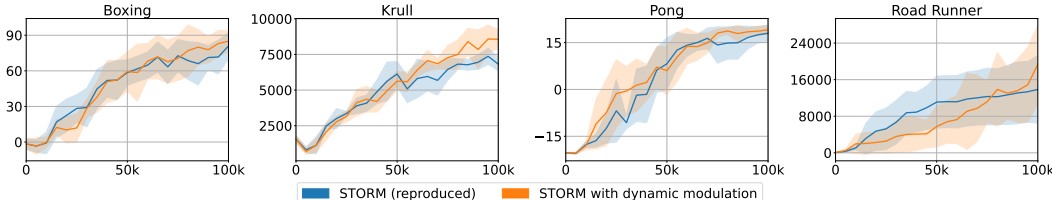

Figure 13: STORM with Dynamic Modulation .

# J Ablations

In the following, we provide additional ablation studies to explore the effect of different components of DyMoDreamer. We conduct all ablations on four games: (i) *Krull*, a game that features multiple levels, with small and numerous dynamic objects, (ii) *Boxing*, a game with a single level and characterized by large and sparse dynamic objects, (iii) *Pong*, a game with a single level and characterized by small and sparse dynamic objects. (iv) *Road Runner*, a game with both small and large dynamic objects. We use the same hyperparameters as described in Appendix K for all ablations.

## J.1 High-dimensional Stochastic Representations

Scaling to larger models leads to higher data efficiency and episode rewards [62]. For example, in TDMPC2 [3], a larger parameter count leads to improved results in the Meta-World [63] and DeepMind Control (DMC) [64] environments. Therefore, we compare the vanilla DreamerV3 that extends $z_t$ to 48 categories with 48 classes for the sake of fairness.

The result is shown in Figure 14, which reveals that only the dimension added in vanilla DreamerV3 $z_t$ do not achieve the same performance as DyMoDreamer. This once again indicates that dynamic modulation is not equivalent to simply increasing dimensions but rather enriching the information used by agents for decision making.

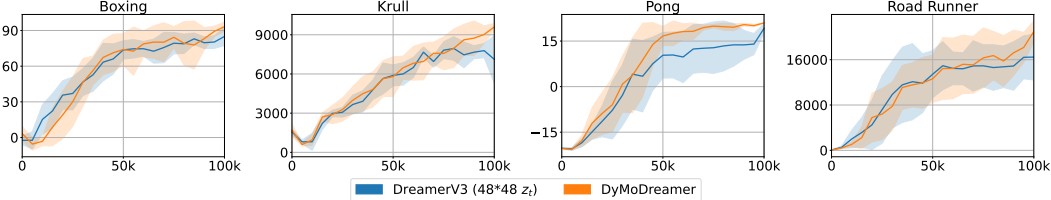

Figure 14: Ablation studies on the high-dimensional DreamerV3.

## J.2 Reconstruction of Differential Observations

To improve the world model's focus on reward-relevant objects, OC-STORM introduces an additional object-centric model, Cutie [55], based on target segmentation. This enhancement significantly improves the baseline STORM's performance. OC-STORM encodes both raw and object-centric observations to obtain stochastic representations and reconstructs these separately during the autoregressive prediction process to train the VAEs. Following a similar approach, we conduct an ablation study on reconstructing differential observations in DyMoDreamer, with results shown in Figure 15. Specifically, we introduce an additional term $\mathcal{L}_{\mathrm{dif}}(\phi) = -\ln p_\phi(o'_t \mid h_t, d_t)$ into the prediction loss function (6). Furthermore, we incorporate a dynamic decoder into (3):

$$\text{Dynamic decoder:} \quad \hat{o}'_t = p_\phi\big(\hat{o}'_t \mid h_t, d_t\big). \tag{21}$$

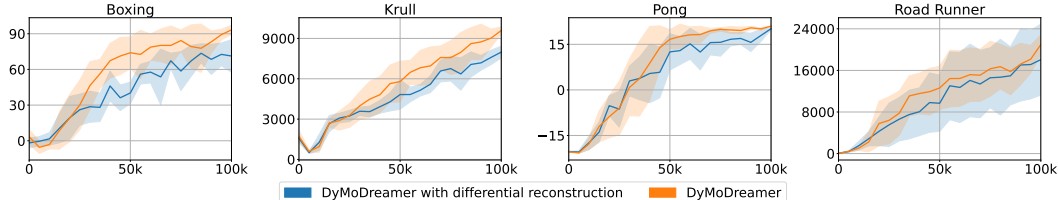

Figure 15: Ablation study on differential observation reconstruction. Adding a dynamic decoder does not improve performance and increases computational cost.

Figure 15 shows that adding a dynamic decoder does not improve DyMoDreamer's performance. Instead, it weakens dynamic modulation's effectiveness and significantly increases computational cost. This is because differential observations are not strictly edge segmentations. Their primary function is to isolate and encode dynamic features of the environment rather than improving its reconstruction.

The substantial improvement observed in the *Pong* environment suggests that when dynamic objects are both small and sparse, differential observations serve as a relatively accurate reconstruction target, allowing the agent to match the performance of pre-trained visual models with a faster speed. Explicitly constraining the dynamic modulators with a decoder may actually degrade performance due to coarse reconstruction targets. In contrast, implicitly training stochastic representations and dynamic modulators together within a single decoder is more efficient and yields better performance. We claim once again that the goal of dynamic modulation is not to improve the reconstruction accuracy, but to enable the agent to leverage richer dynamic information for decision making.

### J.3 Modulation with Latent Difference

We conducted additional experiments using latent-space differencing in place of our dynamic modulation mechanism. The latent difference [30] in MFRL is defined as $\delta_t = z_t - z_{t-1}$, where $z_t$ and $z_{t-1}$ are latent features extracted at consecutive time steps by the same encoder. Similarly, we also experimented with using $\delta_t$ in place of $d_t$ as input to the sequencemodel $f_\phi$. In this setting, the sequence model is changed into:

$$\text{Sequence model:} \quad h_t = f_\phi\big(h_{t-1}, z_{t-1}, \delta_{t-1}, a_{t-1}\big), \tag{22}$$

where the dynamic modulation is retained, coming from $\delta_t$ but not the dynamic modulators $d_t$. Both the computation and encoding of differential observations are removed. The results show that this approach fails to match DyMoDreamer's performance, particularly in environments like *Road Runner*, where the critical objects occupy minimal pixels (e.g., the seed and steel occupying just 1 pixel). This occurs because: (1) The original encoding's precision limits small dynamic feature capture. (2) The categorical distributions in latent space lose fine-grained positional information. (3) Without the explicit dynamic modulation based on differential observations, frame-to-frame differences of small objects become unreliable. DyMoDreamer's strength lies in its hybrid approach combining pixel-level dynamic detection with latent categorical encoding, which preserves critical small-scale dynamics while filtering noise.

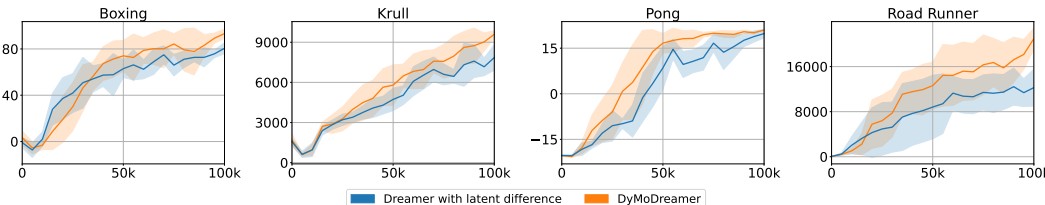

Figure 16: Ablation study on latent difference RSSM.

We emphasize again that DyMoDreamer is fundamentally different from the latent flow, despite both being inspired by the general idea of differencing. DyMoDreamer separates the representation pathways: the dynamic modulator $d_t$ and the stochastic representation $z_t$ are derived from two

different encoders (Equation 1). Specifically, $z_t$ is encoded from the full observations $o_t$, while is encoded from the differential observations $o_t'$ derived from the frame-differencing and processed by a distinct dynamic encoder. Due to the characteristics of CNNs, the encoding of differential observations is not equal to the difference between stochastic representations:

$$z_t - z_{t-1} \neq d_t \sim q_\phi\big(d_t \mid h_t, o_t'\big), \tag{23}$$

where the dynamic modulation mechanism allows DyMoDreamer to retain subtle yet decision-critical dynamic cues that might be overlooked by $z_t$ and $z_{t-1}$ alone.

## J.4 DyMoDreamer with Harmonious Loss

An intriguing question is whether the inclusion of additional loss terms or adjustments to existing ones during the training of the world model impacts its effectiveness. The answer is unequivocally yes: even without introducing new loss terms, modifying the weights of existing ones can significantly influence the model's performance [44]. Recent advances, such as HarmonyDream [44], highlight the benefits of adaptively adjusting the weights of different loss terms during training. By employing a harmonious loss framework [65], this approach achieves performance superior to that of the vanilla DreamerV3. In this section we provide a detailed discussion on the impact of incorporating harmonious loss within DyMoDreamer, demonstrating its potential for improved training efficiency and effectiveness. Adjusting the coefficients of different loss terms has the potential to significantly enhance the performance of world-model-based reinforcement learning algorithms [44]. For DyMoDreamer, compared to DreamerV3, we add three additional loss terms during the training of the world model, further complicating the challenge of tuning the initial loss coefficients. To address this, we adopt a harmonious loss strategy inspired by HarmonyDream [44], dynamically adjusting the dominance between the observation and the reward modeling during world model training process. In this section, we explore the impact of this approach on the performance of DyMoDreamer.

To simplify the presentation and avoid excessive complexity, we consider the vanilla end-to-end loss as below:

$$
\begin{aligned}
\mathcal{L}(\phi) = &\ \omega_{\text{img}}\mathcal{L}_{\text{rec}}(\phi) + \omega_{\text{rew}}\mathcal{L}_{\text{rew}}(\phi) + \omega_{\text{con}}\mathcal{L}_{\text{con}}(\phi) + \omega_{\text{dyn}}\mathcal{L}_{\text{dyn}}(\phi) \\
&+ \omega_{\text{rep}}\mathcal{L}_{\text{rep}}(\phi) + \omega_{\text{reg}}\mathcal{L}_{\text{reg}}(\phi).
\end{aligned} \tag{24}
$$

Then the harmonious loss is constructed as:

$$\mathcal{L}_h(\phi, \omega_i) = \sum \frac{1}{\omega_i}\mathcal{L}_i + \ln(1 + \omega_i) \tag{25}$$

where $i \in \{\text{img}, \text{rew}, \text{con} \, \text{dyn}, \text{rep}, \text{reg}\}$, and specifically, inspired by HarmonyDreamer, we recombine the dynamics and representation losses into $\mathcal{L}_d(\phi)$ as follows:

$$\mathcal{L}_d(\phi) = \alpha\mathcal{L}_{\text{dyn}}(\phi) + (1 - \alpha)\mathcal{L}_{\text{rep}}(\phi). \tag{26}$$

And $\alpha$ is the KL balancing coefficient predefined the same as (5). The reason why we use $\ln(1 + \omega_i)$ as regularization terms is that $\omega_i$ is parameterized as $\omega_i = \exp(\omega_i) > 0$ to optimize parameters $\omega_i$ free of sign constraint. Since a loss item with small values, such as the reward loss, can lead to extremely large coefficient $1/\omega_i \approx L - 1 \gg 1$, which potentially hurt training stability. The derivation of the harmonious loss scale is as follows.

To minimize $\mathbb{E}[\mathcal{L}_h(\phi, \omega_i)]$, we force the the partial derivative w.r.t. $\omega_i$ to 0:

$$
\begin{aligned}
\nabla_\omega \mathbb{E}[\mathcal{L}_h(\phi, \omega)] &= \nabla_\omega\Big(\frac{1}{\omega}\mathbb{E}[\mathcal{L}] + \ln(1 + \omega)\Big) = -\frac{1}{\omega^2}\mathbb{E}[\mathcal{L}] + \frac{1}{1 + \omega} \\
\omega^* &= \frac{\mathbb{E}[\mathcal{L}] + \sqrt{\mathbb{E}[\mathcal{L}]^2 + 4\mathbb{E}[\mathcal{L}]}}{2}.
\end{aligned} \tag{27}
$$

Therefore the learnable loss weight, in the rectified harmonious loss, approximates the analytic loss weight:

$$\frac{1}{\omega^*} = \frac{2}{\mathbb{E}[\mathcal{L}] + \sqrt{\mathbb{E}[\mathcal{L}]^2 + 4\mathbb{E}[\mathcal{L}]}} \tag{28}$$

and equivalently, the harmonized loss scale is:

$$\mathbb{E}\Big[\frac{\mathcal{L}}{\omega^*}\Big] = \frac{2}{1 + \sqrt{1 + \frac{4}{\mathbb{E}[\mathcal{L}]}}} < 1, \tag{29}$$

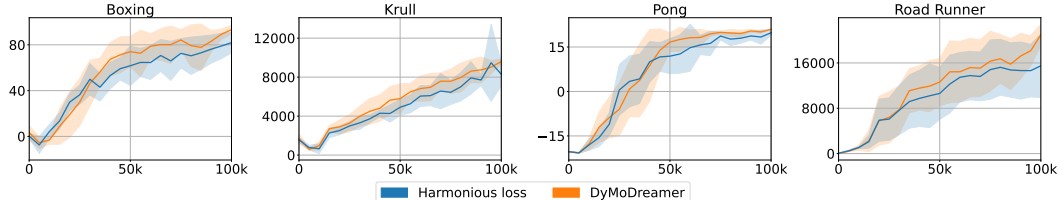

Figure 17: Ablation studies on DyMoDreamer with harmonious loss.

and add the regularization term $\ln(1 + \omega_i)$ results in the $4/\mathbb{E}[\mathcal{L}]$ in the $\sqrt{1 + 4/\mathbb{E}[\mathcal{L}]}$ term, which prevents the loss weight from getting extremely large when faced with a small $\mathbb{E}[\mathcal{L}]$. Regrettably, Figure 17 indicates that while the harmonious loss achieves performance comparable to fixed parameters only in the *Pong* task, but falls short in *Krull*, *Boxing* and *Road Runner* tasks. A potential explanation for this discrepancy is the increased complexity introduced by additional loss terms. The learning rate inherited from the original settings may no longer be optimal, as it appears overly aggressive for adapting the harmonious coefficients.

### J.5 Dimensions of the Dynamic Modulation

The proportion of dynamic objects varies significantly across tasks, as evidenced by the differential mask rate. For instance, in the *Pong* task, the average mask rate (AMR) for a single episode reaches as high as 98.5%, reflecting the minimal volume of dynamic components. In contrast, the AMRs for single episodes in *Boxing* and *Krull* tasks are 94.3% and 83.1%, respectively, due to the larger proportion of dynamic elements in these environments. Consequently, the different dimensionality of $d_t$ can significantly impact the performance of world models. For instance, OC-STORM treats the number of objects as a hyperparameter, and manually adjustis it for different tasks. This task-specific modification results in varying dimensions of the stochastic encoding output by the visual model, which partially enhances OC-STORM's performance [21]. Furthermore, the relative size between $z_t$ and $d_t$ could also influences performance. Increasing the dimension of $d_t$ effectively enlarges the latent variable space, potentially introducing redundant information that could hinder the agent's decision-making process.

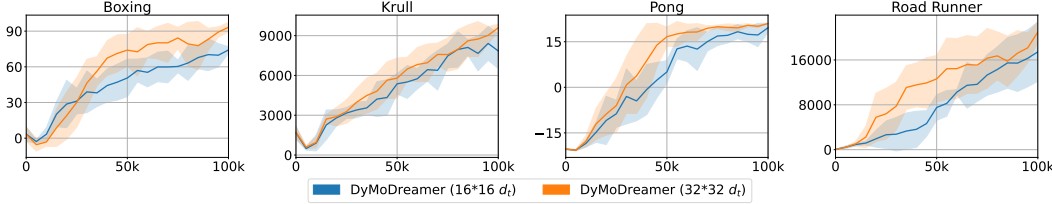

Figure 18: Ablation studies on the dimension of dynamic modulation.

The results in Figure 18 indicate that using $16 \times 16$ classified dynamic modulators does not fully encode the dynamic information from the differential observations. In other words, low-dimensional dynamic modulation fails to accurately integrate dynamic information into the RSSM, thereby not enriching the decision-making process.

### J.6 Longer Difference Interval

Although using single-frame differences already yields substantial performance improvements, we include an ablation study on the difference interval $k$ in this section. Specifically, we set $k = 3$ to evaluate DymoDreamer, and present the results in Figure 19.

The results show that using a longer differencing interval encourages the world model to focus on longer-term changes rather than transient noise or local perturbations. This often leads to better performance in smoother environments, but may fail to capture critical dynamics occurring in intermediate frames, such as the rapid punching in *Boxing*.

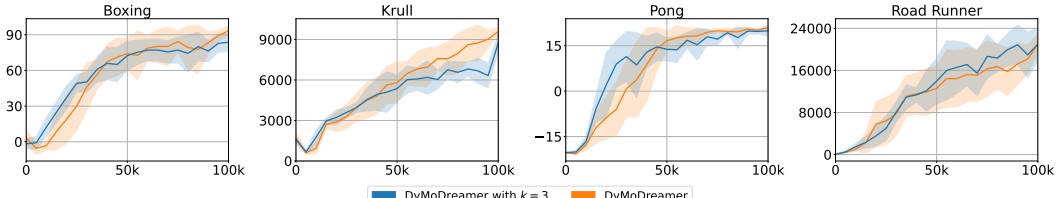

Figure 19: Ablation studies on $k = 3$.

## J.7 Dimensionality of the Stochastic Representations

We additionally examined how the dimensionality of the original stochastic representations $z_t$ influences overall performance. The results are summarized in Figure 20. Our findings indicate that a low-dimensional $z_t$ leads to degraded performance. As we discussed in Appendix H, $z_t$ does not merely encode static content. Visualization suggests that $z_t$ tends to capture dynamic elements in the environment that are beyond the agent's control, whereas the dynamic modulators $d_t$ focuses more on agent-controllable dynamics (despite being decoded from all dynamic components). Therefore, reducing the dimensionality of $z_t$ limits the model's capacity to capture rich environmental dynamics, leading to a notable drop in overall performance.

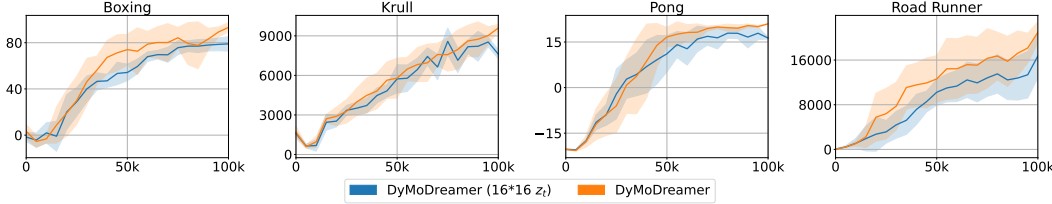

Figure 20: Ablation studies on the dimension of stochastic representations.

# K    Hyperparameters

Table 7 details the hyperparameters of the optimization and environment, as well as hyperparameters shared by multiple components. The environment will provide a "done" signal when losing a life, but will continue running until the actual reset occurs.

Table 7: Hyperparameters

| Name | Value |
| --- | --- |
| **General** | |
| Batch size | 16 |
| Batch length | 64 |
| Activation | LayerNorm + SiLU |
| Optimizer | Adam |
| **World Model** | |
| Learning rate | $10^{-4}$ |
| Adam epsilon | $10^{-8}$ |
| Gradient clipping | 1000 |
| **Actor Critic** | |
| Imagination horizon | 15 |
| Discount horizon | 333 |
| Return lambda | 0.95 |
| Critic EMA decay | 0.98 |
| Critic EMA regularizer | 1 |
| Actor entropy scale | $3 \times 10^{-4}$ |
| Learning rate | $3 \times 10^{-5}$ |
| Adam epsilon | $10^{-5}$ |
| Gradient clipping | 100 |

