# OpenReview forum: "DyMoDreamer: World Modeling with Dynamic Modulation"
_NeurIPS.cc/2025/Conference — NeurIPS 2025 poster_

### Official Review · Reviewer_Whue · 2025-06-24

**Clarity:** 4
**Significance:** 2
**Originality:** 3
**Rating:** 4
**Confidence:** 3

**Summary:**

This paper presents a novel mechanism that leverages pixel-level inter-frame differences to generate stochastic categorical modulators, which dynamically influence the recurrent state-space model to focus on task-relevant motion cues . This approach sidesteps the need for high-fidelity reconstructions or external object masks, resulting in a lightweight world model that boosts sample efficiency in high-dimensional environments .

**Questions:**

1.	Could you report some results on environments like Procgen to demonstrate out-of-distribution generalisation?
2.	How does the modulation branch behave in photorealistic or cluttered environments?
3.	Please provide an aggregated ablation table so that the effect of each architectural component is immediately clear.

**Ethical Concerns:**

["NO or VERY MINOR ethics concerns only"]

**Final Justification:**

Questions were answered.

**Limitations:**

Yes

**Quality:**

3

**Strengths And Weaknesses:**

Strengths:
1. Elegant and Lightweight Design.
2. Significant Empirical Improvements on Atari 100k

Weaknesses:
– All reported experiments use identical train/test distributions. Without a held-out-level benchmark such as Procgen, it is hard to tell whether the dynamic-modulation mask over-fits to textures seen during training.
– Simple frame differencing can break down under motion blur, camera shake or illumination changes that typify photorealistic simulations or real-world robotics. The current evidence, limited to low-resolution games, leaves the method’s applicability to such settings uncertain.
– The paper motivates pixel-space modulation with intuitive arguments and an empirical ablation, yet offers no information-theoretic or representational analysis explaining why pixel differencing consistently outperforms latent differencing.
– Ablation studies are presented only as per-game evaluation curves for a handful of Atari titles. Without an aggregated table, readers cannot gauge the average or worst-case contribution of each component.
- While the authors motivate a focus on world-model variants, readers will inevitably ask how the approach stacks up against strong model-free and search-based methods under the same data budget.

---

> ### Author Rebuttal · Authors · 2025-07-30
>
> We thank the reviewer for the thoughtful and constructive feedback. Below we address each of the raised concerns in detail.
>
> **1. Results on the held-out-level benchmark**
>
> We agree that generalization to unseen levels is crucial for assessing model robustness. Our current evaluation benchmarks (Atari 100k, DMC, and Crafter) are aligned with prior work focusing on sample efficiency and reward-driven dynamics modeling. Notably, the Crafter benchmark shares important characteristics with Procgen, such as procedurally generated content, stochastic transitions, and distributional shifts across episodes. Crafter already introduces distributional shift through its procedurally generated world, in which agent-centric motion relative to a dynamic background challenges traditional frame-differencing approaches. DyMoDreamer's superior performance on Crafter (9.5% improvement over DreamerV3) offers preliminary evidence that our dynamic modulation generalizes beyond static backgrounds.
>
> Moreover, conducting a full evaluation across Procgen benchmark with DreamerV3's settings (the hard difficulty setting and the
> unlimited level set) would require approximately **16 (games) $\times$ 16.1 (days per task) $\times$ 5 (seeds) = 1288 (A100 GPU days)**. Since reducing the evaluation checkpoint to shorter environment steps often results in underdeveloped policies to Procgen benchmark, we conducted additional experiments on the DeepMind Lab benchmark, which is an egocentric 3D game platform designed for RL and contains difficult tasks (e.g.,maze navigation and objects exploration). Considering that our main goal is to improve the sample efficiency of RL algorithms, we shorted the evaluation checkpoint at 10M environment steps (100M steps in the DreamerV3 paper) and focused on several representative tasks with relatively dense reward signals, which are more likely to exhibit performance improvements within 10M steps [1] to assess DyMoDreamer’s performance. To ensure the fairness and quality of our results, we also reproduced DreamerV3 results using the official code and configurations, the corresponding results are reported below. In response to concerns about out-of-distribution generalisation, we ensure that all evaluations are conducted with randomized environment seeds.
>
> | Task | DyMoDreamer | DreamerV3 (reproduced) |
> |:--------------|------:|------:|
> | Explore Goal Locations Small |**164.5**|156.6|
> | Explore Object Rewards Many |  **28.2**  |  24.1 |
> | Lasertag Three Opponents Small |  **8.3**  | 8.2 |
> | Rooms Keys Doors Puzzle| 24.1 |  **26.2**  |
> | Rooms Watermaze |  **22.0**  |  21.2  |
>
> DyMoDreamer outperforms the reproduced DreamerV3 baseline on 4 out of 5 tasks, showcasing strong performance particularly in environments with localized motion-dependent rewards (e.g., Explore Goal Locations Small, Explore Object Rewards Many) and dynamic visual input (e.g., Rooms Watermaze). In contrast, Rooms Keys Doors Puzzle—which prioritizes long-horizon memory and discrete, symbolic planning over visual motion cues—favors DreamerV3, aligning with our analysis of environment suitability.
>
> While the differential observations in DyMoDreamer may not perfectly isolate task-relevant entities, they significantly reduce decision-irrelevant background clutter (e.g., the sky background). This suppression enhances exploration efficiency by directing the agent’s attention toward regions more likely to yield reward. Furthermore, as we discussed in lines 95-98, differential observations are not merely a visual processing technique, they also serve as a direct embedding of temporal information into the model’s representation, enriching its capacity to reason over dynamic sequences. Collectively, these results indicate that DyMoDreamer’s dynamic modulation mechanism also provides robust and generalizable benefits in visually complex and moderately egocentric domains, though its advantage may diminish in settings dominated by abstract reasoning or delayed, non-perceptual dependencies.
>
> [1] Beattie C, et al. DeepMind Lab. arXiv:1612.03801, 2016.
>
> **2. Modulation in photorealistic or cluttered environments**
>
> While our current experiments are conducted in controlled simulators, the architectural design of DyMoDreamer emphasizes temporal change detection rather than reliance on appearance or segmentation priors. This property lends itself naturally to generalization in photorealistic or cluttered scenes (e.g., the Rooms Watermaze task in DeepMind Lab). Our default differencing mechanism employs a binary mask with a fixed threshold to highlight DyMoDreamer's generality and compatibility with a wide range of differencing strategies. In practice, DyMoDreamer can be readily extended to incorporate more robust alternatives, such as logical differencing to mitigate flickering backgrounds and exponential moving average differencing to suppress noise, as detailed in Appendix I.5. More importantly, the architecture is fully compatible with advanced dynamic extractors, including optical flow and attention-based motion segmentation modules, enabling a flexible and extensible framework suitable for complex visual scenes.
>
> **3. Benefits of pixel differencing over latent differencing**
>
> Similar to most prior work on world models, we provide both intuitive reasoning and experimental evidence for pixel-space differencing in Section 2.3. Specifically, pixel differencing maintains spatial precision for small but critical features (e.g., 1-pixel objects in Pong) that may be lost in latent encoding due to compression. Moreover, our ablation studies in Appendix I.3 confirms that latent differencing ($\delta_t = z\_t − z\_{t−1}$) underperforms pixel-based modulation, particularly when small motions are not captured by $z\_t$ in the experimental tasks.
>
> **4. Aggregated ablation table**
>
> Thank you for pointing this out. First, we would like to clarify that, as discussed in lines 294–297, the four Atari tasks selected for ablation were chosen because they exhibit distinct dynamic characteristics (e.g., sparse motion, and small/large object dynamics). These settings are closely aligned with the design goals of DyMoDreamer and allow for efficient and targeted evaluation of each architectural component’s contribution to dynamic modulation, since conducting all ablation studies across 26 Atari tasks are resource-intensive and infeasible effort given our time and funding budget at present.
>
> In the final version, we will include an aggregated ablation table summarizing the performance of different ablation studies across the four Atari games (Boxing, Pong, Krull, Road Runner). A preliminary version of the table is:
>
>
> | Ablation Variant                   | Boxing | Krull  | Pong  | Road Runner |
> |-----------------------------------|--------|--------|-------|-------------|
> | DyMoDreamer      | **93.6** | **9624.8** | **20.9**  | **20971.8**  |
> | Removing dynamic modulation            | 80  | 7969   | 18.5  | 12536  |
> | Removing  $\mathcal{L}_{reg}$ | 90  | 8961  | 20  | 20918   |
> | High-dimensional $z\_t$ (48x48)        | 76   | 7325   | 18.2  | 16320       |
> | With differential reconstruction  | 71  | 7895   | 19.9  | 17323      |
> | Latent difference  |81   | 7855  | 20  | 12266   |
> | Low-dimensional $d\_t$ (16x16)  |73  | 7423  | 19  | 17465   |
>
> These results reinforce that the dynamic modulation mechanism is crucial for robust performance across diverse game types, and that simply appending difference images without architectural integration leads to suboptimal results (Appendix I).
>
> **5.Comparison with MFRL and search-based methods**
>
> Compared to model-free approaches, DyMoDreamer exhibits significantly higher sample efficiency and achieves substantially better performance on a number of Atari tasks (compared with BBF). This highlights the strength of world model–based reasoning in low-data regimes. Furthermore, unlike model-free methods that often require millions of environment interactions to converge, DyMoDreamer reaches competitive or superior performance within a constrained budget, showcasing its practical advantages in sample-limited settings. Search-based methods such as MuZero requires 8 TPUs for training per task, whereas our JAX implementation of DyMoDreamer achieves competitive performance with just 1 GPU for 5.5 hours, demonstrating significantly better computational efficiency while maintaining the benefits of world model learning.
>
> We hope that our responses have adequately addressed all your concerns. Please feel free to let us know if any further clarification is needed.

---

### Official Review · Reviewer_zbN2 · 2025-06-30

**Clarity:** 4
**Significance:** 3
**Originality:** 3
**Rating:** 4
**Confidence:** 4

**Summary:**

This paper proposes DyMoDreamer, a new model-based RL (MBRL) algorithm with a dynamics modulation mechanism to extract and utilize dynamic features of the observation. The model builds on DreamerV3 and combines a masking function to extract dynamical parts of the observation and a regularization loss to motivate the model to capture dynamic changes of the environment better. In experiments, DyMoDreamer outperforms other MBRL algorithms, as the model can attend better to parts of the observation with dynamical changes.

**Questions:**

- Related to Weakness 2, you mention that dynamic parts of the observation are often correlated with rewards (line 175). Given this, have you considered conditioning the dynamics modulation mechanism on the agent's actions? It seems this could be a possible direction to help the model focus on extracting parts of the image that are task-related as well?
- While I agree that both dynamic and static parts of the observations are important for task solving (line 182), could you elaborate on the design choice not to fully disentangle these two information channels in the model's representation (e.g., by creating separate static (1-M(o_t)) ⋅ o_t and dynamic M(o_t) ⋅ o_t pathways)?
- I think one of the motivation behind this paper was to improve the model’s capability to capture dynamics of small, moving objects. To further strengthen this claim beyond the helpful visualizations, have you considered a more quantitative evaluation? For example, using object segmentation or tracking metrics on the extracted dynamic masks could provide a powerful, objective measure of the model's capability in this area.

**Ethical Concerns:**

["NO or VERY MINOR ethics concerns only"]

**Final Justification:**

Overall, my concerns have been largely addressed and my questions answered, and I recommend borderline accept.

**Limitations:**

yes

**Quality:**

3

**Strengths And Weaknesses:**

Strengths:

- The paper proposes a novel MBRL model, DyMoDreamer, that improves capturing dynamical changes in the environment by encoding temporally differential observations. DyMoDreamer can predict small dynamic objects’ movements better, which is also useful for the policy learning as well.
- Quantitative results show DyMoDreamer outperforms other RSSM- and Transformer-based MBRL algorithms on 3 benchmark datasets. Qualitative results show that the differential observations extracted by DyMoDreamer are the objects that dynamically change in the environment.
- The paper is clearly written, well-structured, and includes a thorough set of ablation studies that provide valuable insights into the contributions of the different architectural components. The figures are informative and effectively support the paper's claims.

Weaknesses:

- The experiments are only conducted in allocentric environments. The environments that this paper uses all have a third-view perspective, allowing the model to “see” the agent itself and its relationship with other players/objects. However, in egocentric environments, such as maze navigation, all the pixels will dynamically modulate due to changes in the agent’s perspective. This will likely lead to the model failing to extracting objects dynamics important in solving the task. This raises the question of the applicability of DyMoDreamer in broader contexts.
- As I understand, the main assumption of the paper is that all dynamically changing aspects of the environment is important for solving the task. However, in more realistic environments, there may be more noises such as moving foreground objects or background noises which are not important. As the extraction is conducted using a simple binary function with unlearnable threshold, the model may fail to distinguish the important agent and objects’ changes and noises that change unrelated to the agent’s actions. This raises a question about the model's robustness in more complex, noisy settings.
- The paper has several minor grammatical errors and typos
    - The sentence in line 130 “Considering that …” seems to be grammatically incorrect
    - In line 667, progcess should be process

---

> ### Author Rebuttal · Authors · 2025-07-30
>
> We sincerely thank you for the constructive feedback on our paper. Below, we address each concern raised, and all new experiments conducted in the response will be incorporated into the final version of our manuscript.
>
> **1. Generalization to egocentric environments DeepMind Lab**
>
> To evaluate DyMoDreamer's generalizability, we conducted additional experiments on the DeepMind Lab benchmark, which is an egocentric 3D game platform designed for RL and contains difficult tasks (e.g.,maze navigation and objects exploration). Classical methods (IMPALA and R2D2+) typically require over 1B environment steps to achieve non-trivial reward performance. Since our main goal is to improve the sample efficiency of RL algorithms, we shorted the evaluation checkpoint at 10M environment steps (100M steps in the DreamerV3 paper) and focused on several representative tasks with relatively dense reward signals, which are more likely to exhibit performance improvements within 10M steps [1] to assess DyMoDreamer’s performance. To ensure the fairness and quality of our results, we also reproduced DreamerV3 results using the official code and configurations, the corresponding results are reported below.
>
> | Task | DyMoDreamer | DreamerV3 (reproduced) |
> |:--------------|------:|------:|
> | Explore Goal Locations Small |**164.5**|156.6|
> | Explore Object Rewards Many |  **28.2**  |  24.1 |
> | Lasertag Three Opponents Small |  **8.3**  | 8.2 |
> | Rooms Keys Doors Puzzle| 24.1 |  **26.2**  |
> | Rooms Watermaze |  **22.0**  |  21.2  |
>
> DyMoDreamer outperforms the reproduced DreamerV3 baseline on 4 out of 5 tasks, showcasing strong performance particularly in environments with localized motion-dependent rewards (e.g., Explore Goal Locations Small, Explore Object Rewards Many) and dynamic visual input (e.g., Rooms Watermaze). In contrast, Rooms Keys Doors Puzzle—which prioritizes long-horizon memory and discrete, symbolic planning over visual motion cues—favors DreamerV3, aligning with our analysis of environment suitability.
>
> While the differential observations in DyMoDreamer may not perfectly isolate task-relevant entities, they significantly reduce decision-irrelevant background clutter (e.g., the sky background). This suppression enhances exploration efficiency by directing the agent’s attention toward regions more likely to yield reward. Furthermore, as we discussed in line 95-98, differential observations are not merely a visual processing technique, they also serve as a direct embedding of temporal information into the model’s representation, enriching its capacity to reason over dynamic sequences. Collectively, these results indicate that DyMoDreamer’s dynamic modulation mechanism also provides robust and generalizable benefits in visually complex and moderately egocentric domains, though its advantage may diminish in settings dominated by abstract reasoning or delayed, non-perceptual dependencies.
>
> [1] Beattie C, et al. DeepMind Lab[J]. arXiv:1612.03801, 2016.
>
> **2. Robustness in noisy or dynamic background**
>
> Our default differencing mechanism employs a binary mask with a fixed threshold to highlight DyMoDreamer's generality and compatibility with a wide range of differencing strategies. In practice, DyMoDreamer can be readily extended to incorporate more robust alternatives, such as logical differencing to mitigate flickering backgrounds and exponential moving average differencing to suppress noise, as detailed in Appendix I.5. More importantly, the architecture is fully compatible with advanced dynamic extractors, including optical flow and attention-based motion segmentation modules, enabling a flexible and extensible framework suitable for complex visual scenes. Furthermore, our experiments in DeepMind Lab demonstrate that foreground object motion often conveys egocentric relative dynamics, which can be informative and beneficial for downstream policy learning.
>
> **3. Dynamical modulation and agent actions**
>
> We thank the reviewer for the thoughtful suggestion to condition the dynamic modulation mechanism on the agent’s actions. We agree that this is a promising direction, and in fact, we believe that the current version of DyMoDreamer already implicitly incorporates this idea. As our model is trained end-to-end, the dynamic modulators participate in the temporal inference process ($f\_{\phi}$  in Equation (3)) alongside the agent’s action, contributing to the prediction of future latent states. The world model is optimized jointly via a reconstruction and policy loss on fixed replay sequences, which inherently couples the modulation signal with the action trajectory. While we do not explicitly encode the action $a\_t$ into the computation of $d\_t$ , the action influences the next latent state $h\_t$, which in turn affects the encoding and interpretation of subsequent modulators $d\_t$, since the encoding process of  $d_t$ is $d_t \sim q_\phi\left(d_t \mid h_t,  o'_t\right)$. Thus, DyMoDreamer performs implicit action-conditioned modulation through temporal feedback in the recurrent dynamics model. We provide supporting empirical evidence for this interaction in Appendix H, although we did not explicitly constrain $d_t$ to focus on the part controlled by the agent, it naturally focuses more on agent-controllable dynamics (despite being decoded from all dynamic components).
>
> **4. Partial vs full dynamics-static disentanglement**
>
> We intentionally avoid explicitly separating dynamic and static components using hard static masking $(1-M(o\_t) \cdot o\_t)$ for the following reasons.
>
> First, our differential observations are derived from frame-differencing followed by dilated convolutions, but not from sophisticated edge-aware segmentation techniques. As a result, the resulting dynamic masks $M(o\_t)$ highlight motion-correlated regions rather than clearly delineated object boundaries. The edges of these masks may not align with semantic object contours, and emphasizing such boundaries through hard segmentation may overemphasize potentially irrelevant edge pixels while neglecting the truly decision-relevant patterns within the masked regions $M(o\_t) \cdot o\_t$.
>
> Moreover, since the differenced observation includes not only the changed pixels but also their local neighborhoods (as we metioned in lines 131-134), directly enforcing a binary split could disrupt the integrity of reward-relevant static features. For instance, partial inclusion of an important static object across both streams could prevent either $z\_t$ or $d\_t$ from capturing it in full, ultimately degrading decision quality. Our joint encoding approach circumvents this issue by allowing the model to learn a soft, task-driven decomposition guided by reconstruction and policy optimization signals.
>
> **5. Quantitative evaluation of dynamic feature modeling**
>
> Standard RL benchmarks typically lack precise real-time annotations, and more complex environments such as DMLab introduce additional visual challenges including stage-wise transitions and shifting viewpoints. As a result, there are no established segmentation or tracking metrics readily applicable for evaluating dynamics modeling within RL settings. In this work, we rely on task performance, imagination accuracy, and qualitative visualizations (Fig. 3, Appendix F) to assess the model's capacity for dynamic feature modeling. We believe these provide sufficiently comprehensive and task-relevant evidence in the absence of standardized evaluation protocols.
>
> **6. Grammatical errors and typos**
>
> We gratefully thank you for the careful check on our manuscript, and we will revise the manuscript accordingly:
>
> $\bullet$ Line 130: In fact, very few pixels may exceed the threshold $\epsilon$ after being processed by $D(\cdot)$.
>
> $\bullet$ Line 667: This emergent specialization suggests an natural segregation within the model’s representational architecture like the denoised Markov decision process [59], since we do not explicitly assign reconstruction targets to different decoders.
>
> We hope we have addressed all your raised points. If there are anything unclear, we would be happy to provide further clarifications.

---

> > ### Author Response · Authors · 2025-08-05
> >
> > We sincerely appreciate the time and effort you have dedicated to reviewing our submission. We would like to respectfully follow up to inquire whether there are any remaining concerns or points in our rebuttal that would benefit from further clarification. We would be more than happy to provide additional information if needed.

---

> > > ### Comment · Reviewer_zbN2 · 2025-08-06
> > > **Re: Rebuttal**
> > >
> > > I thank the authors for addressing my points. While most of my concerns have been addressed, I have an additional question:
> > >
> > > While I understand that the main goal is sample efficiency, I am concerned that the comparison at 10M steps may not be sufficient to claim that DyMoDreamer "outperforms DreamerV3." As the original DreamerV3 paper illustrates, the model's performance on DMLab is still actively improving at this stage. I think a further analysis on DMLab is needed for the final verdict about how well DyMoDreamer can handle allocentric environments. Also, I would like to know about the Average Mask Rate for these tasks as well.

---

> ### Author Response · Authors · 2025-08-06
>
> We sincerely thank the reviewer for their thoughtful feedback and follow-up question. To further clarify our design choices, we elaborate below on the motivation behind our current experimental setup and provide additional details regarding the average mask rate.
>
> **Regarding the 10M-steps comparison with DreamerV3**
>
> We appreciate the concern that DyMoDreamer and DreamerV3's performance on DMLab may continue to improve beyond 10M steps. However, we would like to respectfully point out that the original DreamerV3 paper itself does not always choose fully converged tasks, especially in complex benchmarks such as DMLab. In fact, experiments in DreamerV3 are also conducted under fixed step budgets (e.g., 100M for DM Lab), rather than until absolute convergence. Our evaluation protocol closely follows this precedent.
>
> Importantly, DreamerV3 and DyMoDreamer are specifically designed to enhance sample efficiency, thus the performance in the early training phase is a central metric of interest. Moreover, the successful results of DyMoDreamer on DMLab under the 10M-steps setting demonstrate that our dynamic modulation mechanism generalizes well not only to allocentric control tasks, but also to egocentric visual environments such as DMLab. This is also the motivation behind our inclusion of the additional DMLab experiments and the cross-paradigm applicability suggests that the learned dynamics-aware attention is not limited to one particular observation structure, but can flexibly adapt to different types of visual input. Our experiments demonstrate that DyMoDreamer exhibits a significantly faster learning curve compared to DreamerV3 under the same 10M-step budget. Importantly, the slope of DyMoDreamer‘s curve beyond 8M steps remains comparable to DreamerV3, suggesting that the performance advantage is unlikely to vanish in the later stage. As we consider the practical constraints on training time and compute resources (since conducting a full evaluation across DM Lab benchmark would require approximately **30 (tasks) $\times$ 2.9 (days per task) $\times$ 5 (seeds) = 435 (A100 GPU days)**, we believe that evaluating DyMoDreamer under the 10M interactions constraint is both meaningful and justified since faster convergence is itself a valuable property for world models. This evaluation strategy aligns with the growing trend in the literature where many works (e.g. STORM [1], TWM [2], and MWM [3]) choose Atari100k or Meta-world with limited interactions as the sole benchmark to emphasize early-stage learning performance.
>
> [1] Zhang W, et al. Storm: Efficient stochastic transformer based world models for reinforcement learning[J]. Advances in Neural Information Processing Systems, 2023, 36: 27147-27166.
>
> [2] Robine J, et al. Transformer-based World Models Are Happy With 100k Interactions[C]. The Eleventh International Conference on Learning Representations.
>
> [3] Seo Y, Hafner D, et al. Masked world models for visual control[C]. International Conference on Robot Learning. PMLR, 2023: 1332-1344.
>
> **Regarding the average mask rate**
>
> We have now included the average mask activation rates for the DMLab experiments, as shown in the table below. We would like to emphasize that the value of the mask rate does not directly determine performance. Rather, the mask serves as a mechanism to highlight regions of dynamic change, allowing the model to modulate its representation accordingly. Even relatively sparse masks (e.g., the Bank Heist task in Atari) can lead to meaningful improvements, as they focus learning on temporally salient features without introducing excessive noise. Our empirical results indicate that performance gains are more influenced by the effectiveness of the modulation than the density of the mask itself.
>
> |Tasks| Average Mask Rate (10M) |
> |:--------------|:------:|
> | Explore Goal Locations Small|7.8%|
> | Explore Object Rewards Many|  11.2%  |
> | Lasertag Three Opponents Small|  20.7%  |
> |Rooms Keys Doors Puzzle|32.3%|
> | Rooms Watermaze |  10.3% |
>
> We hope this response clarifies the rationale behind our evaluation protocol, and we sincerely appreciate your time and consideration in reviewing our additional clarifications.

---

> ### Author Response · Authors · 2025-08-09
>
> To further address the reviewer’s concern, we have extended the experiments to 25M environment steps as of the time of this comment. Preliminary evaluations indicate that DyMoDreamer continues to maintain its earlier advantage (we proposed in the rebuttal) over DreamerV3. Notably, DyMoDreamer has already surpassed DreamerV3’s performance in the Rooms Watermaze environment at a converged level, and the corresponding results are reported in the table below. We are continuing to run the experiments to 100M steps and will include these results in the final version of the paper, which we expect will further substantiate DyMoDreamer’s ability to handle allocentric environments while preserving strong sample efficiency.
>
> |Task | DyMoDreamer (25M)| DreamerV3 (25M)|
> |:--------------|------:|------:|
> | Explore Goal Locations Small|**323.0**|321.1|
> | Explore Object Rewards Many|  **48.2**  |  39.9 |
> | Lasertag Three Opponents Small|  **14.3**  |12.7 |
> |Rooms Keys Doors Puzzle| 27.3|  **29.1**  |
> | Rooms Watermaze |  **28.1**  |  27.3  |

---

> > ### Comment · Reviewer_zbN2 · 2025-08-09
> > **Re: Rebuttal**
> >
> > Thank you for your detailed reply as well as updated experiments. I will adjust my score accordingly.

---

> > > ### Author Response · Authors · 2025-08-09
> > >
> > > We sincerely thank the reviewer for the thoughtful follow-up and for considering our updated experiments in the evaluation. We truly appreciate your willingness to adjust the score, your constructive feedback throughout the review process, and your recognition of our supplementary experimental setup.

---

### Official Review · Reviewer_2wbS · 2025-07-02

**Clarity:** 3
**Significance:** 3
**Originality:** 3
**Rating:** 5
**Confidence:** 5

**Summary:**

The paper introduces DyMoDreamer, an extension of DreamerV3 that aims to decouple dynamic elements from static components in pixel-based observations. Dynamic modulation is achieved by (1) including differential observations as additional inputs, encoded in a dedicated latent space, and (2) incorporating a loss term that emphasizes better prediction of inter-frame variation. The method demonstrates strong performance on three benchmarks (Atari100k, DeepMind Control Suite, and Crafter) achieving or surpassing state-of-the-art results.

**Questions:**

1. Dimensionality of $z$: In the provided examples $z$ seems to mostly capture static backgrounds. Did the authors experiment with the dimensionality of the static latent $z_t$? Have you considered using a learned but fixed/static $z$ as a baseline?
2. Clarification on the "Removing Dynamic Modulation" Ablation: Could you clarify what is meant by this ablation? Is the decoder predicting both the original and differential images jointly? Are these inputs concatenated, or are separate decoder heads used?
3. Baseline with $\mathcal{L}_\mathrm{reg}$: Is there an ablation experiment comparing against a version of DreamerV3 that incorporates only the differential divergence regularization loss (without modulation)? Comparing against such a baseline would help disentangle the contribution of the modulation mechanism from the loss term.
4. Applicability Beyond Images: DreamerV3 has also been used for environments with state-based inputs. Could DyMoDreamer extend to such cases? Do the authors believe that dynamic modulation would provide similar benefits in non-visual settings?
5. Broader Impact of Architectural Biases: Does separating dynamic and static latents introduce inductive biases that may hurt performance in environments where this assumption doesn’t hold (e.g., rapidly changing backgrounds)?


Minor comments:
- Line 27: “A key approach to addressing this challenge is the world model [...]. This framework combines a VAE with a recurrent neural network (RNN) and uses evolutionary strategies to optimize the policy within a latent space.” The first sentence reads as a general statement about world models, but the second appears to describe the specific approach of Ha & Schmidhuber. It would be helpful to clarify which parts refer to the general framework versus this specific implementation.
- Equations 2 & 9: The indices $i,h,w,c$ should be introduced in natural language before or alongside the equation. Clarifying what these dimensions represent (e.g., batch index, height, width, channels) would improve readability and help readers unfamiliar with the notation.

**Ethical Concerns:**

["NO or VERY MINOR ethics concerns only"]

**Final Justification:**

The dynamic modulation mechanism introduced in this paper leads to strong performance across several benchmarks. While some reviewers raised concerns about the narrow scope of environments (e.g., egocentric views), the authors addressed this by including additional evaluations in DMLab. My own concerns regarding missing baselines (regularization term, applicability beyond pixel-based inputs) were addressed through new ablations. The authors also demonstrated how dynamic modulation can be applied to another world model architecture (STORM), supporting the generality of the approach, albeit on a model structurally similar to Dreamer. They additionally clarified the paper’s limitations, which I believe should be stated more clearly in the final version. Overall, my concerns have been addressed and my questions answered, so I recommend acceptance. However, as the paper is mainly of interest to the model-based RL community, I did not assign a higher score.

**Limitations:**

Limitations are not clearly addressed. The paper would benefit from a dedicated discussion of limitations. For example, a brief analysis of scenarios where dynamic/static separation could fail would strengthen the contribution.

**Quality:**

3

**Strengths And Weaknesses:**

Strengths:
- an original extension of the Dreamer line of work, emphasizing the modeling of dynamic structure in visual inputs.
- strong empirical performance across a diverse set of benchmarks, which showcases the method's significance to the model-based RL community.
- the paper is generally well-written, and the figures and illustrations clearly communicate key ideas.

Weaknesses:
- certain ablation details are either missing or under-explained, e.g., the role of the dimensionality of the static latent component ($z_t$) and the inclusion/exclusion of regularization terms like $\mathcal{L}_\mathrm{reg}$ from Dreamer.
- the method is closely tied to Dreamer-style architectures and pixel-based inputs; its generality or applicability to other world models or state-based inputs is unclear, which limits broader impact.
- the related work section focuses exclusively on world models, even though the approach appears to draw from modulated ODEs and latent flow methods. Including a discussion of these inspirations would help clarify the method’s novelty.

---

> ### Author Rebuttal · Authors · 2025-07-30
>
> We sincerely thank the reviewer for the insightful comments. Below, we address each point in detail, and we will incorporate clarifications and additional results in the final version.
>
> **1. Dimensionality of stochastic representations $z\_t$**
>
> We analyzed the effect of dynamic modulators' dimensionality in Appendix I.5, and additionally examined how the dimensionality of the original stochastic representations $z\_t$ influences overall performance. The results are summarized in the table below. Our findings indicate that a low-dimensional $z\_t$ also leads to degraded performance. As we discussed in Appendix H, $z\_t$ does not merely encode static content. Visualization suggests that $z\_t$ tends to capture dynamic elements in the environment that are beyond the agent’s control, whereas the dynamic modulators $d\_t$ focuses more on agent-controllable dynamics (despite being decoded from all dynamic components). Therefore, reducing the dimensionality of $z\_t$ limits the model’s capacity to capture rich environmental dynamics, leading to a notable drop in overall performance. This is why we don't use a completely static fixed $z$. In fact, in our early experiments, we have explored a variant where a fixed static vector $z$ replaced $z\_t$ to encode/decode only the background (achieved by averaging over the batch length dimension), while only the dynamic modulators $d\_t$ were used to drive the sequence model. Under this configuration, the performance was comparable to or worse than vanilla DreamerV3, as the static background can shift over the course of an episode (e.g., the level transitions in Krull). A fully fixed  $z$ lacks the flexibility to adapt to such stage-wise changes in the static background, ultimately hindering representation quality and decision-making.
>
> | Score        | Boxing  | Krull | Pong | RoadRunner |
> |:-------------|:------:|:-----------:|:---------:|:--------:|
> | DyMoDreamer  | **93.6** | **9624** | **21** | **20971** |
> | DyMoDreamer with $16 \times 16$ $z\_t$| 79.0 |   7612   |  16.3   |  16732  |
>
> **2. Clarification on the “Removing Dynamic Modulation”**
>
> In this ablation, a single decoder is used to reconstruct only the original observation $o\_t$, without reconstructing the differential observations $o'\_t$. The decoder receives a concatenation of $z\_t$ (encoded from  $o\_t$) and $d\_t$ (encoded from  $o'\_t$) as input, which is fed through a shared decoding head (the reconstruction process is formulated the same as Equation (1)). In this configuration, $d\_t$ is not explicitly embedded into the RSSM as in our dynamic modulation design.
>
> **3. Baseline with $\mathcal{L}\_{reg}$**
>
> We thank the reviewer for this insightful comment. In response, we implemented a baseline using DreamerV3 with the same divergence regularization term $\mathcal{L}\_{reg}$ as DyMoDreamer but without dynamic modulation. This variant showed minor improvement over DreamerV3 but underperformed compared to DyMoDreamer (in the following table). The results demonstrate that the differential divergence regularization acts as a complementary supervisory signal that works in concert with reconstruction objectives, and contributes improvement through its unique ability to constrain the temporal consistency in the reconstruction process of vanilla DreamerV3.
>
> | Score        | Boxing  | Krull | Pong | RoadRunner |
> |:-------------|:------:|:-----------:|:---------:|:--------:|
> |DreamerV3 (reproduced)  |80 | 7834 | 18 |**15248**|
> |DreamerV3 with $\mathcal{L}\_{reg}$ |**82** |**8240**|**18.1** |13649|
>
> **4. Applicability beyond images and generality to other world models**
>
> To evaluate the applicability of DyMoDreamer in state-based environments, we conducted experiments on DeepMind Proprio—a proprioceptive control benchmark where observations consist of structured state vectors such as velocity and position. In this setting, observation differencing naturally reduces to state differencing, with temporal dynamics being more explicitly encoded. Consequently, differential modulation provides similar benefits in non-visual settings, since such differences also carry meaningful physical semantics similar to the differential observations—for instance, differential positions implicitly encode velocity, while the differential velocities reflect acceleration.
>
> | Score        | Acrobot Swingup  | Cheetah Run | Finger Spin |
> |:-------------|:------:|:-----------:|:---------:|
> |DyMoDreamer |**225**|**625**|**932**|
> |DreamerV3|134 |614|**931** |
>
> To further validate the effectiveness of our dynamic modulation mechanism, we integrated it into STORM [1], a recent efficient transformer-based world model. As shown in the results below, this integration also leads to clear performance gain.
>
> | Score        | Boxing  | Krull | Pong | RoadRunner |
> |:-------------|:------:|:-----------:|:---------:|:--------:|
> |STORM (reproduced)  |81 | 6824 | 18 |13866|
> |STORM with dynamic modulation |**85** |**8563**|**19.1** |**19337**|
>
> [1] Zhang W, et al. STORM: Efficient stochastic transformer based world models for reinforcement learning. Advances in Neural Information Processing Systems, 2023, 36: 27147-27166.
>
> **5. Broader impact of architectural biases**
>
> A naive separation of dynamic and static components may introduce inductive biases in certain environments. To mitigate such issues (dynamic backgrounds or environmental noise), we provide several alternative differencing strategies in Appendix G, including multi-frame logical differencing and moving average differencing. Importantly, the dynamic modulation mechanism is highly compatible with a wide range of advanced, difference-based computer vision techniques. This extensibility suggests that DyMoDreamer possesses strong generalization potential, as differencing is a foundational principle in many visual detection methods.
>
> **6. Discussion of modulated ODEs and latent flow**
>
> While modulated ODEs also employ modulation operators to enhance the temporal modeling in neural ODEs, our approach differs in several key aspects. In modulated ODEs, modulators are **time-invariant** and trained as a global statistic across the entire sequence, whereas DyMoDreamer extracts modulation explicitly from differential observations at each timestep. Additionally, modulated ODEs encode modulators from the full input sequence rather than isolating dynamic cues via differential observations. As a result, modulated ODEs are primarily suited for modeling passive, uncontrolled dynamics, whereas our dynamic modulation framework enables explicit modeling of environment transitions conditioned on agent actions.
>
> The latent flow methods largely focused on **model-free reinforcement learning** and do not introduce dedicated encoders or decoders for the latent flow  $\delta\_t$. The latent flow $\delta\_t$ are not designed to be predictive but serve as auxiliary information injected into decision-making. In contrast, DyMoDreamer explicitly incorporates the dynamic modulators into the reconstruction process to mitigate hallucinations in dynamic regions (as shown in Fig. 3). Moreover, the modulators $d\_t$ are fully predictive and plays a central role in improving the world model’s capacity to model temporally coherent dynamics.
>
> **7. Revision in the manuscript**
>
> We gratefully appreciate your valuable comments and suggestions, and we will revise the manuscript accordingly:
>
> $\bullet$ We will clarify the distinction by rewriting the sentence in line 27 as:
> “A key approach to addressing this challenge is the general framework of world models, which learns compact latent dynamics for planning and decision-making. One early instantiation of this framework, proposed by Ha & Schmidhuber, combines a VAE with an RNN and uses evolutionary strategies to optimize policies in the learned latent space.”
>
> $\bullet$ Equations (2) & (9): We will introduce the indices in natural language before the equations to improve clarity. Specifically, we will indicate the meaning of each index (e.g.,$i$ is the time dimension, $h$ and $w$ are spatial dimensions, and $c$ is the channel dimension) to assist readers unfamiliar with the notation.
>
> We appreciate the reviewer’s thoughtful feedback and hope that our rebuttal has addressed your concerns. Please kindly let us know if further details are required.

---

> > ### Comment · Reviewer_2wbS · 2025-08-04
> > **Thank you and follow-up question**
> >
> > Thank you for the detailed response and the additional experiments. I believe these clarifications and new results clearly strengthen the contribution of the paper.
> > Could you clarify how the STORM world model was modified to integrate dynamic modulation?

---

> > > ### Author Response · Authors · 2025-08-04
> > > **STORM with Dynamic Modulation**
> > >
> > > Thank you for the encouraging feedback and for your interest in how DyMoDreamer extends the core dynamic modulation mechanism. We will elaborate on how dynamic modulation is integrated into STORM from the following three key perspectives: (1) the world model architecture, (2) the end-to-end training objective, and (3) the learning process of agent.
> > >
> > > (1) First, analogous to the improvements made by DyMoDreamer over the DreamerV3 framework, we extend the STORM framework by introducing a dedicated dynamic encoder $d_t \sim q_{\phi}(d_t|o'\_t) = \mathcal{D}\_t$ that processes the differential observations $o'\_t$ (designed in DyMoDeamer) and is separate from the original observation encoder $z_t\sim q_{\phi}(z_t|o_t)=\mathcal{Z}\_t$. The original encoder and the dynamic encoder specifically samples the stochastic representations $z_t$ and the dynamic modulators $d_t$ from the distribution $\mathcal{Z}\_t$ and $\mathcal{D}\_t$, and the decoder then reconstructs the observation using both $z_t$ and $d_t$ by $\hat{o}\_t = p_{\phi}(z_t,d_t) $. Throughout this process, we adhere to STORM's original design choice by not incorporating the hidden state $h_t$ into the encoding path. Furthermore, following the design of STORM, we concatenate the stochastic representations $z_t$, the dynamic modulators $d_t$, and the actions $a_t$  into unified tokens $e_t$ ($e_t = m_{\phi}(z_t, d_t, a_t)$), which is then fed into the Transformer-based sequence model $f_{\phi}$ to produce the hidden states $h_{1:T} = f_{\phi}(e_{1:T})$. Finally, the hidden state is used to predict the stochastic representations' distribution $\hat{\mathcal{Z}}\_{t+1}$, the dynamic modulators' distribution $\hat{\mathcal{D}}\_{t+1}$, the reward $\hat{r}\_{t}$, and the continuation flag $\hat{c}_{t}$ via dedicated predictors (MLPs). This process is similar to the Equation (3) in DyMoDreamer and serves to integrate the modulation mechanism into the sequence model, ensuring that the dynamic modulator remains predictable.
> > >
> > > (2) In the aspect of the end-to-end training objective, STORM does not introduce additional objectives beyond those in DreamerV3. Following a similar strategy to DyMoDreamer, we extend STORM’s training objective by incorporating regularization terms on the dynamic modulators in both the dynamics loss (Equation (7) in DyMoDreamer) and the representation loss (Equation (8) in DyMoDreamer). Additionally, we introduce the differential divergence regularization term  (Equation (10) in DyMoDreamer) into the overall loss function to further encourage temporal consistency in STORM with dynamic modulation.
> > >
> > > (3) The agent’s learning is also solely based on the imagination process facilitated by the world model like DyMoDreamer, and the actor-critic operates on the concentration states $s_t=(h_t,z_t,d_t)$. To initiate the imagination process, a brief contextual trajectory is randomly selected from the replay buffer, and the initial posterior distributions $\mathcal{Z}\_t$ and $\mathcal{D}\_t$ are computed. During inference, rather than sampling directly from the posterior distributions $\mathcal{Z}\_t$ and $\mathcal{D}\_t$,  $z_t$ and $d_t$ are sampled from the prior distribution $\hat{\mathcal{Z}}\_{t+1}$ and $\hat{\mathcal{D}}\_{t+1}$. The remaining setup follows STORM which is the same as DreamerV3
> > >
> > > In summary, the integration of dynamic modulation into STORM is structurally and functionally aligned with DyMoDreamer’s design principles. These modifications allow the world model to capture reward-relevant dynamics more effectively while maintaining architectural efficiency and training stability.
> > >
> > > We hope this explanation clarifies how dynamic modulation is integrated into STORM. Please let us know if there are any remaining questions or aspects that could benefit from further clarification.

---

> > > > ### Comment · Reviewer_2wbS · 2025-08-05
> > > > **Response to Rebuttal**
> > > >
> > > > Thank you for the detailed response. My questions have been answered and I recommend acceptance of the paper.

---

> > > > > ### Author Response · Authors · 2025-08-05
> > > > >
> > > > > We are glad that our clarifications and additional results helped to address your concerns, and we sincerely appreciate you taking the time to re-evaluate the paper and recognizing our contributions.

---

### Official Review · Reviewer_z8ms · 2025-07-23

**Clarity:** 3
**Significance:** 3
**Originality:** 2
**Rating:** 4
**Confidence:** 4

**Summary:**

DyMoDreamer introduces a low‑cost extension and high efficiency to Dreamer‑style (Hafner et al) world models, introducing a scalable way to incoporate dynamic object information without any explicit priors.

The paper argues that the recurrent state‑space model (RSSM) of Dreamer V3 with an extra categorical "dynamics modulator" $d_t$,  that is inferred from frame‑difference masks and concatenated to the usual stochastic latent z_t. This forces the model to attend to motion cues without expensive vision pre‑training and with minimal extra compute. The paper shows state‑of‑the‑art scores on Atari 100k, DeepMind Control (pixels) and Crafter, plus ablations about their design choices.

**Questions:**

The paper's "latent‑difference" ablation simply subtracts the quantised categorical codes: z_t - z_{t-1} and feeds the raw vector into the RSSM, with no re‑encoding, normalisation, or spatial structuring. That choice likely under‑represents what a proper latent‑difference approach can do. It would be good to see more detailed discussion about this.

Additionally, it would be good to see more discussion or experimantation with more explicit approaches which directly model object dynamics (with physics or 3D priors for example).

**Ethical Concerns:**

["NO or VERY MINOR ethics concerns only"]

**Limitations:**

yes

**Quality:**

3

**Strengths And Weaknesses:**

Strengths:

I think this paper tackles an important problem of introducing object level information without any explicit tracking, bounding boxes etc, and is inspired from how human think about *differences* between scenes and not just scene features.

Additionally, the paper is clear, well‑explained method. The paper is easy to follow and the algorithmic additions are minimal yet well-motivated. It is also provides strong empirical evidence to support the claims. Across three benchmarks, DyMoDreamer improves on previous state of the art results, such as Dreamer V3. Moreover, The ablations are quite informative for example, removing the modulator or differential divergence regularizer drops Atari performance. Finally, the writing is clear and there are many details which enhance reproducibility, i. e, implementation choices, loss terms, and training schedules are spelled out.

Weakness:
In my opinion, there is limited evaluation beyond fixed‑camera games. Unlike DIAMOND (which at least simulates a first‑person CS:GO environment) and Dreamer V3 (which learns on 30 egocentric DMLab tasks), DyMoDreamer reports results only on fixed‑view benchmarks. Therefore, it remains unclear whether its frame‑difference modulation can handle rapid viewpoint changes, parallax, or background motion typical of true ego‑centric domains.

---

> ### Author Rebuttal · Authors · 2025-07-29
>
> We sincerely thank the reviewer for the comprehensive evaluation and respond to the raised concerns below, specifically regarding the generalizability of DyMoDreamer to egocentric domains, the rationale behind the latent-difference ablation design, and the necessity of discussing explicit object dynamics or 3D modeling approaches. All additional experiments referenced in this rebuttal will be included in the final version of the paper.
>
> **1. Evaluation on egocentric DM Lab tasks**
>
> In response to the concerns about the evaluation on more explicitly egocentric environments, we conducted additional experiments on the DeepMind Lab benchmark to further validate DyMoDreamer's effectiveness under rapid viewpoint changes. To the best of our knowledge, among existing world model literature, only DreamerV3 has been evaluated on this benchmark. Unfortunately, we were unable to locate a readily integrable pip package or a pre-configured Dockerfile for streamlined deployment, therefore setting up the environment and managing dependencies via Bazel incurred substantial overhead during our supplementary experimentation phase. Moreover, conducting a full evaluation across DM Lab benchmark would require approximately **30 (tasks) $\times$ 2.9 (days per task) $\times$ 5 (seeds) = 435 (A100 GPU days)**. To make the evaluation tractable, we shorted the evaluation checkpoint at 10M environment steps (100M steps in the DreamerV3 paper) and focused on several representative tasks with relatively dense reward signals, which are more likely to exhibit performance improvements within 10M steps [1] to assess DyMoDreamer’s performance. To ensure the fairness and quality of our results, we also reproduced DreamerV3 results using the official code and configurations, the corresponding results are reported below.
>
> | Task | DyMoDreamer | DreamerV3 (reproduced) |
> |:--------------|------:|------:|
> | Explore Goal Locations Small |**164.5**|156.6|
> | Explore Object Rewards Many |  **28.2**  |  24.1 |
> | Lasertag Three Opponents Small |  **8.3**  | 8.2 |
> | Rooms Keys Doors Puzzle| 24.1 |  **26.2**  |
> | Rooms Watermaze |  **22.0**  |  21.2  |
>
> DyMoDreamer outperforms the baseline on 4 out of 5 tasks, demonstrating competitive performance particularly on tasks with localized motion rewards (e.g., Explore Goal Locations Small, Explore Object Rewards Many) and dynamic visual input (e.g., Rooms Watermaze). However, in Rooms Keys Doors Puzzle, which emphasizes long-horizon memory and discrete planning over motion cues, DreamerV3 shows a slight advantage, consistent with our analysis of task suitability. In such settings, although the differential observations may not isolate only the task-relevant objects, they nonetheless effectively suppress large portions of decision-irrelevant background. This facilitates more efficient exploration by directing the agent’s attention toward reward-associated regions of the scene. Furthermore, as we discussed in lines 95-98, differential observations are not merely a visual processing technique, they also serve as an embedding of temporal information into the model’s representation, enriching its capacity to reason over dynamic sequences. These results suggest that DyMoDreamer’s dynamic modulation mechanism offers generalizable benefits in visually rich and moderately egocentric environments, though it may be less effective in domains requiring complex, abstract reasoning or long-term symbolic memory.
>
> [1] Beattie C, et al. DeepMind Lab. arXiv:1612.03801, 2016.
>
> **2. Latent-difference ablation design**
>
> We thank the reviewer for the valuable remark. Our initial choice, directly subtracting latent states to construct $\delta_t=z_t-z_{t-1}$ without re-encoding or spatial structuring was intentional to establish a conservative baseline for comparison with DyMoDreamer. To ensure a more rigorous and fair ablation, we conducted an additional ablation experiment where $\delta_t$ was not only fed into the sequence model ($h\_t = f\_{\phi}(h\_{t-1}, z\_{t-1}, \delta\_{t-1}, a\_{t-1})$), but also participated in the reconstruction ($ p\_{\phi}(\hat{o\_t}|h\_t,z\_t,\delta\_{t})$, $ p\_{\phi}(\hat{r\_t}|h\_t,z\_t,\delta\_{t})$, and $ p\_{\phi}(\hat{c\_t}|h\_t,z\_t,\delta\_{t})$) and the policy learning processes ($s\_t= \lbrace h\_t,z\_t,\delta\_{t}\rbrace$). In this configuration, the usage of the latent flow $\delta_t$ aligns with its formulation in the original work, except that it additionally participates in the reconstruction process. As reported in the table below, substituting $d\_t$ with $\delta\_t$ does not lead to improved performance, further underscoring the unique effectiveness of our proposed dynamic modulation signal. As discussed in Section 2.3 (lines 187–194) and Appendix I.3, we empirically confirmed that differencing in latent space will fail to capture subtle but critical motion signals (e.g., 1-pixel ball in Pong) due to quantization and precision loss, ultimately leading to degraded performance.
>
> | Score        | Boxing  | Krull | Pong | RoadRunner |
> |:-------------|:------:|:-----------:|:---------:|:--------:|
> | DyMoDreamer  | **93.6** | **9624** | **21** | **20971** |
> |Replace $d\_t$ with $\delta\_t$|  82.1  |   7461   |  15   |  16932  |
>
> **3. Discussion on modeling object dynamics**
>
> Explicitly modeling object dynamics using 3D priors or physics engines is an important direction. Our work emphasizes an orthogonal strategy: enhancing dynamic feature capture without explicit object segmentation or prior assumptions will improve the world model's performance. To more explicitly explore the incorporation of object dynamics through prior modeling, we conducted an additional experiment in which the differenced observations were replaced with a pretrained computer vision module "Cutie". Cutie, informed by prior knowledge, is capable of directly extracting key objects from the scene after pretraining. In this setting, the dynamics captured by $d\_t$ (where $d\_t$ are encoded from the Cutie's output in this case) corresponds exclusively to the motion of key objects, with other information effectively excluded. While this approach does yield performance gains comparable to those of DyMoDreamer, it introduces additional training overhead through reliance on supervised pretraining, and incurs a higher computational cost than our lightweight modulation mechanism.The results of this experiment are presented below.
>
> | Score        | Boxing  | Krull | Pong | RoadRunner |
> |:-------------|:------:|:-----------:|:---------:|:--------:|
> | DyMoDreamer  | **93.6** | **9624** | **21** | 20971 |
> |DyMoDreamer with Cutie|  92.1  |   9443  |  21   | **21852** |
>
> We hope our response has resolved the concerns, and please let us know if any additional clarifications are required.

---

> > ### Author Response · Authors · 2025-08-05
> >
> > We truly appreciate the time and effort you’ve devoted to reviewing our paper. We just wanted to gently follow up to see if there are any aspects of our rebuttal that remain unclear or would benefit from further elaboration. We would be glad to provide any additional clarification or supporting details if that would be helpful.

---

### Note · Authors · 2025-08-12

We sincerely thank the reviewers, AC, and SAC for the time, all constructive comments, and the fruitful and engaging discussions. We deeply appreciate the thoughtful feedback and the recognition of our motivation and experimental design, which have helped us further clarify and enhance our contributions.

In this paper, we propose DyMoDreamer, a novel world model architecture for model-based RL that incorporates a dynamics modulation mechanism. This mechanism leverages time-variant modulators to allocate the model's capacity to different regions of the visual input, emphasizing dynamic areas while down-weighting static or decision-irrelevant ones. DyMoDreamer has achieved superior sample efficiency and sustained performance gains in both fixed-view allocentric settings and egocentric environments across Atari100k, DeepMind Visual Control, Crafter and DeepMind Lab. Moreover, the dynamics modulation mechanism is general and can be integrated into other world model frameworks to enhance their focus on task-relevant dynamics, potentially improving the representation learning and the policy performance. At the same time, DyMoDreamer can be extended with more robust differencing strategies, including logical differencing to reduce flickering backgrounds and moving average differencing to suppress noise, which further improve its effectiveness in complex visual scenes.

We will incorporate the valuable suggestions from the review process into the final version to further improve clarity, completeness, and scope. We are grateful for the community's input and for the opportunity to present our work.

---

### Decision · Program_Chairs · 2025-09-17

**Decision:**

Accept (poster)

**Comment:**

This paper introduces DyMoDreamer, a model-based RL algorithm that explicitly models dynamic objects via inter-frame differencing and dynamic modulation. The method is simple yet effective, achieving state-of-the-art results on Atari100K, DMC, and Crafter. The paper is well-motivated and clearly written.

The main limitation is that experiments are confined to relatively simple simulated domains, where its applicability to realistic scenes and tasks questionable.

The rebuttal addressed reviewer concerns satisfactorily.

Overall, I recommend acceptance: the contribution is novel, well-presented, and empirically strong.